# MITIGATING CONVERSATIONAL INERTIA IN MULTI-TURN AGENTS THROUGH CONTEXT BIAS CALIBRATION

## ABSTRACT

Large language models excel as few-shot learners when provided with appropriate demonstrations, yet this strength becomes problematic in multi-turn agent scenarios where excessive mimicry of previous interactions undermines agentic exploration. We identify the root cause as *conversational inertia*—a phenomenon where models exhibit strong diagonal attention to previous assistant responses, creating imitation bias that constrains exploration. This phenomenon manifests prominently at moderate context lengths (e.g., 4K tokens) and worsens with longer conversations, explaining why agent performance degrades well before reaching the model's context limits. Through attention analysis, we find that models increasingly focus on previous responses while attention to task instructions shows marginal change, disrupting the exploration-exploitation balance for agents. We propose Context Bias Calibration as a unified framework to mitigate inertia. Our approach operates through two complementary mechanisms: clip context periodically clears interaction history, and Context Preference Learning that calibrates model preferences to favor responses generated with shorter contexts over those from longer contexts, using their own outputs without environment rewards. Experimental results across eight diverse environments demonstrate that Context Bias Calibration Framework reduces conversational inertia and achieves performance improvements.

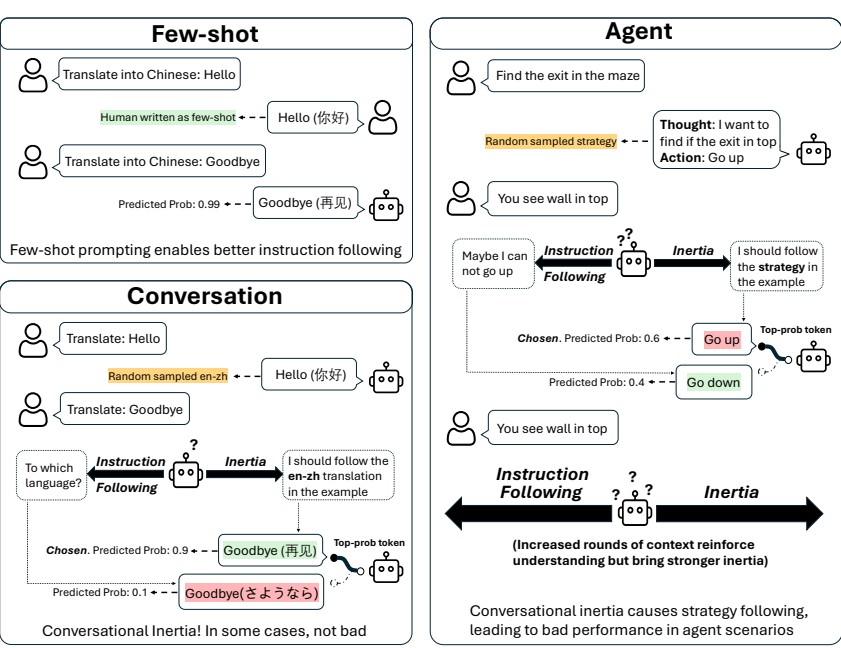

Figure 1: Illustration of LLMs incorrectly transferring few-shot capabilities to agent domains.

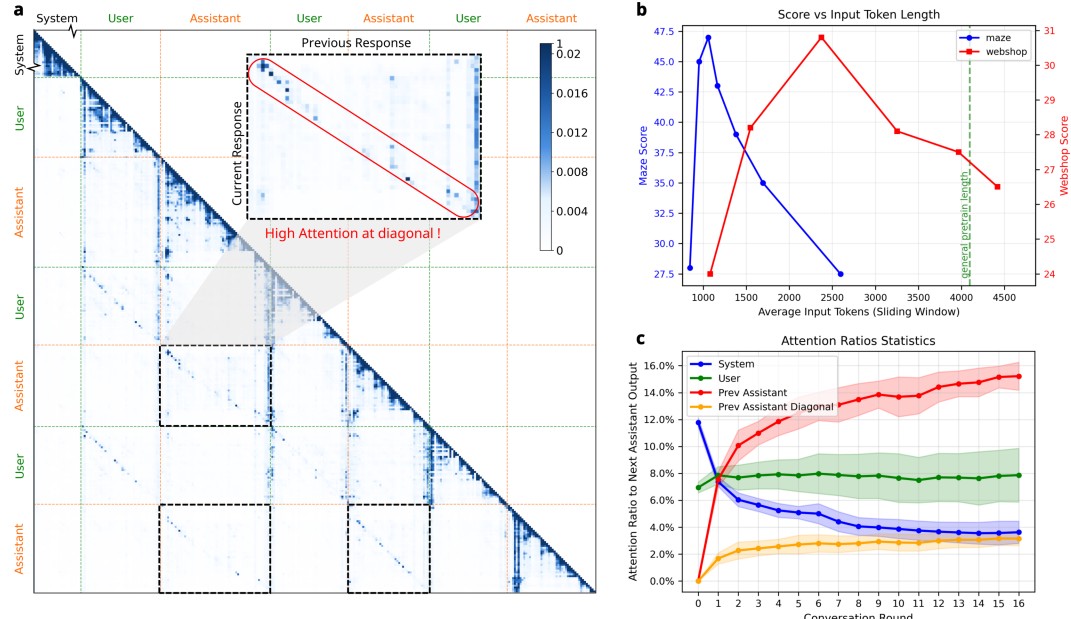

Figure 2: Analysis of conversational inertia in multi-turn dialogue agents. **(a)** Attention visualization in maze environment. Despite generating diverse responses, the model exhibits strong diagonal attention to previous assistant outputs, revealing token-to-token correspondence that creates conversational inertia. For other environments, see Appendix Q. **(b)** Performance degradation with average context length sent to the agent. Agent capabilities decline well before reaching model context limits, which conventional "long context degradation" explanations fail to account for these phenomena. **(c)** Quantitative analysis of each part of attention as context increases. It reveals monotonic growth in attention to previous assistant responses, while attention allocation to user inputs remains stable, confirming the context bias hypothesis. Detailed analysis is presented in Section 3.7.

# 1 INTRODUCTION

The emergence of Large Language Models (LLMs) has revolutionized autonomous task completion, with multi-turn dialogue agents becoming increasingly prevalent following the introduction of paradigms like ReAct (Yao et al., 2023b). These agents interact with environments through iterative observation-action cycles, accumulating context over extended episodes to solve complex tasks such as web navigation (Yao et al., 2022), embodied AI (Shridhar et al., 2021), and so on.

However, a critical challenge emerges as dialogue length increases: agent performance degrades significantly even when context remains well within the model's capacity. Recent empirical analysis (Hsieh et al., 2024a; Liu et al., 2024) attributes this degradation to two factors: approaching context window limits due to length generalization issues, and the inherent difficulty of attention mechanisms in processing long sequences. While these explanations seem intuitive, they fail to account for performance drops observed at moderate context lengths (e.g., 4K tokens) that are far below modern models' capabilities (Hong et al., 2025), as illustrated in Figure 2(b).

To understand this phenomenon, we conduct a detailed analysis of attention patterns in multi-turn agent interactions. Our investigation reveals a previously unidentified issue: *conversational inertia*. Through attention matrix visualization in Figure 2(a), we discover that models exhibit strong diagonal correspondence when attending to previous assistant responses. Specifically, the $i$-th token in the current response disproportionately attends to tokens around the $i$-th position in previous assistant outputs, creating an imitation bias that accumulates errors and constrains exploration for agents.

This attention pattern has profound implications for agent imitation behavior: Models tend to replicate previous response patterns rather than adapting to new environmental feedback, leading to suboptimal decision-making (Laban et al., 2025). Furthermore, as context length increases, attention to

system prompts becomes reduced, causing models to rely more heavily on few-shot learning from interaction history rather than following task-specific instructions. Our empirical analysis confirms that this phenomenon represents imitation of homogeneous information rather than simple length-related degradation. As evidence, attention to previous assistant responses grows monotonically with context length, while attention to user inputs remains relatively stable, as shown in Figure 2(c).

Given this context bias phenomenon, a natural question arises: *can we mitigate conversational inertia through context control mechanisms?* We address this challenge through our context bias calibration approach, which aims to mitigate imitation bias arising from context and conversation history through two complementary mechanisms: **First**, a training-free clip context method that periodically clears interaction history to reset inertial patterns while preserving task-relevant information. Our clip context method creates breaks at conversation turn boundaries by simply clipping several oldest contexts to $L$ rounds every $H$ rounds. This design prevents error propagation by eliminating accumulated inertial patterns. Additionally, this clip design naturally enables KV cache optimization through its append-only structure between clip operations. While sliding windows (Shinn et al., 2023; Deng et al., 2024) still allow old errors to influence new actions through continuous context overlap, and because each step prefix is changed cannot use KV cache speedup. **Second**, to strengthen bias calibration at the model level, we develop Context Preference Learning method that fine-tunes only 0.4% of parameters using long-short context preference pairs. Our method teaches models to prefer actions generated from shorter contexts, which exhibit weaker conversational inertia, over actions from longer contexts. Such preference learning enables models to resist inertial degradation, requiring no ground truth environment rewards or expert demonstrations.

Extensive evaluation across eight diverse environments demonstrates that our approach outperforms strong baselines. Experimental results show that both clip context and Context Preference Learning methods directly mitigate conversational inertia, reducing diagonal attention patterns by noticeable margins.

The key contributions of this work are: (1) identification and detailed analysis of conversational inertia as a cause of multi-turn agent performance degradation, (2) a training-free clip context method that breaks inertial patterns while providing computational efficiency gains, and (3) a Context Preference Learning method that reduces context bias without requiring environment supervision.

## 2 CONTEXT BIAS CALIBRATION FRAMEWORK

### 2.1 TRAINING-FREE CLIP CONTEXT METHOD

To address conversational inertia and enable agents to break out of error loops, motivated by the finding in Figure 2(c) that diagonal attention is reduced under shorter history, we propose a unified perspective that encompasses existing context management approaches and introduces our clip-based attention mechanism, as illustrated in Figure 3.

Let $\mathcal{C}_t = \{r_1, r_2, \ldots, r_t\}$ denote the conversation context in turn $t$, where each round $r_i$ consists of user input $u_i$ and assistant response $a_i$. For round-wise attention control, we define the attention mask $M_t \in \{0, 1\}^{|\mathcal{C}_t| \times |\mathcal{C}_t|}$ that determines which historical rounds are accessible during the generation of current response. We formulate three primary context control strategies through their attention masking patterns.

**Full Context** Attention method preserves the complete conversation history $\mathcal{C}_t$ at each turn, allowing unrestricted access to all previous rounds without using any masking.

**Window Context** Attention method maintains only the most recent $W$ rounds in $\mathcal{C}_t$, masking older interaction rounds.

**Clip Context (Ours)** Attention method implements periodic attention masking. When the clearing threshold $H$ is reached, the context mask is reset to contain only the $L$ most recent rounds.

To optimize computational efficiency, we transform the attention masking formulation into a context trimming approach. Rather than maintaining full context with selective masking, we directly manipulate the conversation history:

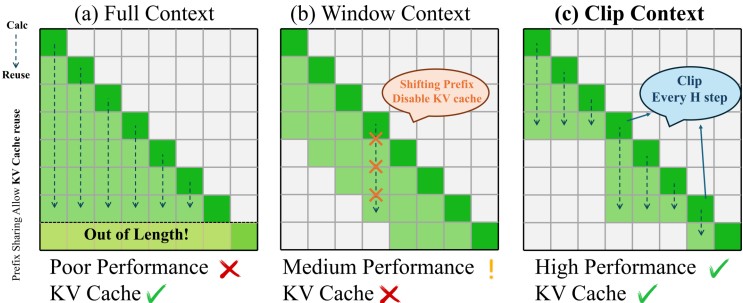

Figure 3: Comparison of attention masking strategies for context control. **(a)** Full Context Attention retains complete context but suffers from conversational inertia and theoretically has context capacity limit problems. **(b)** Window Context Attention maintains recent context but cannot leverage KV cache due to shifting boundaries. **(c)** Clip Context Attention (our method) periodically clears context while enabling KV cache optimization.

$$\mathcal{C}_t^{\text{trimmed}} = \begin{cases} \{r_{\max(1,t-L+1)}, \ldots, r_t\} & \text{if } |\mathcal{C}_{t-1}^{\text{trimmed}}| + 1 = H \\ \mathcal{C}_{t-1}^{\text{trimmed}} \cup \{r_t\} & \text{otherwise} \end{cases} \tag{1}$$

Our training-free clip context approach offers four key advantages: (1) **Inertia Breaking**: Periodic context clearing mitigates positional correspondence patterns that cause conversational inertia. As shown in Figure 2(b), when rounds in context are low, there is much less diagonal attention (see Section 3.7). (2) **Balance exploration and exploitation**: Our clip context method periodically alternates between short and long contexts, enabling better exploration-exploitation balance: short contexts reduce inertia for exploration, long contexts utilize interaction history for exploitation. (3) **Prevent Error Accumulation**: Unlike the window context approach that always keeps the newest $W$ rounds in context, our method periodically creates breaks in context, thus preventing error accumulation across interaction rounds (see Section B). (4) **Computational Efficiency**: KV cache compatibility reduces inference overhead compared to sliding window approaches that cannot leverage prefix caching. Our clip context method achieves approximately $W\times$ (window size) speedup in prefill computation compared to sliding window approaches. See Appendix E for why window context cannot use KV cache. Detailed theoretical computational complexity analysis is provided in Appendix F.

## 2.2 CONTEXT PREFERENCE LEARNING METHOD

Inspired by the performance degradation patterns in Figure 2(b) and the diagonal attention growth patterns in Figure 2(c), we propose a Context Preference Learning method. Our key insight is that for identical contextual states, actions generated with longer input contexts exhibit stronger conversational inertia effects compared to those generated with shorter contexts. We leverage this observation to construct **strong-weak inertia preference** pairs, without requiring any environment-provided reward signals or expert demonstrations. Using this data, we teach the model to prefer lower-inertia actions.

As illustrated in the left panel of Figure 4, for each state we generate two actions:

- $a^{\text{long}}$ using the full conversation history (longer context)
- $a^{\text{short}}$ using only recent context (shorter context)

Then, we execute the long-context generated action $a^{\text{long}}$ in the environment to obtain the next observation, which is then used to synchronously update both the long and short context trajectories. This design ensures that the dataset's input context represents lower-quality trajectories, while the preferred chosen actions correspond to higher-quality decisions, encouraging the model to break out of loops in lower-quality input contexts and generate better actions.

The preference dataset uses the shorter context as training input for both chosen and rejected actions. This design prevents the model from learning to rely primarily on recent history while ensuring

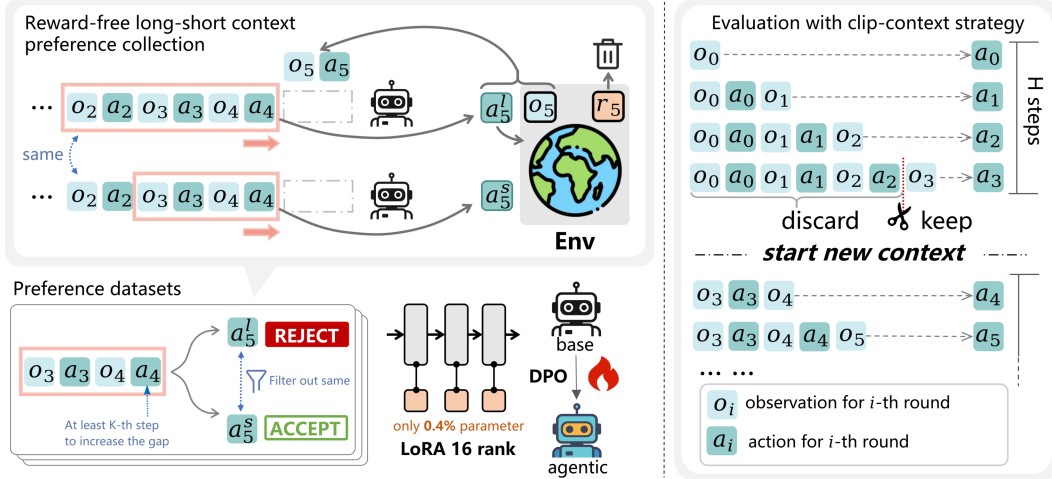

Figure 4: Context Preference Learning method. **Left**: Preference data collection and DPO (Rafailov et al., 2023) training. Our method does not need environment rewards ($r_5$) to generate preference pairs. **Right**: Evaluation showing how models applies clip context method during inference.

robust preference understanding even with limited context exposure. To enhance data quality, we implement two filtering strategies: (1) minimum context margin requiring at least $K$ steps before sampling to ensure sufficient context disparity, and (2) action diversity filtering excluding pairs where $a_t^{\text{long}} = a_t^{\text{short}}$ to focus on cases where context length meaningfully impacts decision-making. The DPO loss function becomes:

$$\mathcal{L}_{\text{DPO}} = -\mathbb{E}\left[\log \sigma \left(\beta \log \frac{\pi_\theta(a_t^{\text{short}}|\mathcal{C}_t^{\text{short}})}{\pi_{\text{ref}}(a_t^{\text{short}}|\mathcal{C}_t^{\text{short}})} - \beta \log \frac{\pi_\theta(a_t^{\text{long}}|\mathcal{C}_t^{\text{short}})}{\pi_{\text{ref}}(a_t^{\text{long}}|\mathcal{C}_t^{\text{short}})}\right)\right] \tag{2}$$

where $\mathcal{C}_t^{\text{short}}$ is consistently used as input context for both preference options. Then, we employ parameter-efficient LoRA (Hu et al., 2022) fine-tuning for DPO training. Detailed training configuration and hyperparameters are provided in Appendix D.1. We also ablate some LoRA hyperparameters in Appendix D.2

## 3 EXPERIMENTS

### 3.1 ENVIRONMENTS

We evaluate our approach across eight environments, covering embodied AI (Shridhar et al., 2021), web interaction (Yao et al., 2022), reasoning-dependent strategic games (Guertler et al., 2025), navigation tasks (Abdulhai et al., 2023), and crafting scenarios (Prasad et al., 2023). We use Agent-Gym (Xi et al., 2024) suite implementation. We also evaluate in Deep Research scenarios based on BrowseComp (Wei et al., 2025), using ReSum (Wu et al., 2025) codebase. For detailed environment specifications and experimental configurations, refer to Appendix N.

### 3.2 IMPLEMENTATION DETAILS

We evaluate our training-free clip context methods using three language models: Qwen3-8B (w/o thinking) (Yang et al., 2025), Llama3.1-8B-Instruct (AI @ Meta, 2024), and GPT-4o-mini. All experiments use temperature 0.8 enabling free exploration and more stable results, rather than temperature 0, which may not adequately reflect the exploration-exploitation balance for agent environments. We use Clip-12 and Window-6 by default, as they have the same average input to agents.

Table 1: Cross-model performance comparison of clip context and Context Preference Learning methods across environments. Success rates (%) are reported. Window-6 and Clip-12 have equivalent average input lengths. Sum-12 extends Clip-12 by adding a summary of previous context generated by Qwen3-8B during clipping. Summarization implementation details are provided in Appendix I.

| Model | Context | MZ | ALF | WS | TC | FL | HM | 2048 | RH | Avg |
|---|---|---|---|---|---|---|---|---|---|---|
| *Qwen3-8B* | | | | | | | | | | |
| Qwen3-8B (actor-only) | Long | 39.0 | 61.0 | 23.3 | 67.5 | 63.5 | 75.1 | 65.6 | 40.3 | 54.4 |
| | Window-5 | 69.0 | 63.2 | 38.7 | 72.8 | 67.7 | 85.8 | 69.0 | 47.7 | 64.2 |
| | Window-6 | 74.0 | **68.2** | 40.0 | 71.3 | 67.5 | 85.3 | 66.9 | 45.8 | 64.9 |
| | Window-7 | 70.0 | 64.5 | 40.6 | 71.3 | 65.0 | 86.0 | 70.2 | 42.8 | 63.8 |
| | **Clip-10** | 82.5 | 65.5 | **45.3** | 81.5 | 66.3 | 85.8 | **72.1** | **51.7** | 68.8 |
| | **Clip-12** | **83.0** | 67.7 | 44.4 | **82.3** | 67.5 | 85.1 | 70.9 | 50.2 | **68.9** |
| | **Clip-14** | 81.5 | 68.2 | 43.2 | 81.8 | 68.1 | 86.6 | 70.1 | 51.4 | 68.9 |
| Qwen3-8B (summary) | Sum-12 | 78.0 | 71.0 | 46.5 | 80.5 | 64.6 | 88.9 | 70.6 | 50.4 | 68.8 |
| Qwen3-8B-RFT | Long | 79.0 | 66.0 | 62.8 | 81.0 | 71.5 | 78.5 | 66.4 | 48.7 | 69.2 |
| | Window-6 | 81.0 | 69.5 | 67.8 | 90.0 | 72.6 | 81.6 | 67.5 | 48.4 | 72.3 |
| | **Clip-12** | **92.0** | **75.0** | **69.6** | **92.0** | 71.8 | 83.3 | 69.5 | **56.3** | **76.2** |
| **Qwen3-8B-DPO (Ours)** | Long | 44.0 | 59.5 | 40.5 | 65.5 | **70.0** | 77.7 | 70.2 | 44.8 | 59.0 |
| | Window-6 | 78.0 | 67.5 | 44.9 | 74.5 | 68.2 | **89.5** | 72.4 | 44.3 | 67.4 |
| | **Clip-12** | **91.5** | **70.3** | **54.9** | **83.0** | 68.4 | 87.5 | **73.0** | **51.1** | **72.5** |
| *Llama3.1-8B-Instruct* | | | | | | | | | | |
| Llama3.1-8B-Instruct | Long | 51.0 | 18.0 | 28.2 | 42.5 | 51.6 | 75.5 | 71.1 | **38.8** | 47.1 |
| | Window-6 | 74.0 | 24.2 | **42.7** | 56.8 | 54.9 | **86.1** | 71.0 | 32.4 | 55.3 |
| | **Clip-12** | **76.0** | **28.5** | 38.7 | **64.5** | **56.0** | 84.3 | **72.1** | 34.1 | **56.8** |
| Llama3.1-8B-DPO | Long | 34.0 | 13.5 | 39.4 | 45.0 | 59.6 | 84.9 | **70.0** | **38.8** | 48.2 |
| | Window-6 | 63.0 | 15.5 | **46.1** | **66.5** | **60.3** | **94.4** | 69.7 | 36.6 | 56.5 |
| | **Clip-12** | **69.0** | **26.0** | 39.4 | 49.0 | 55.4 | 92.9 | 69.9 | 37.6 | **57.4** |
| *Closed Source Models* | | | | | | | | | | |
| GPT-4o-mini | Long | 62.0 | **62.0** | **45.5** | 52.0 | 87.2 | 96.8 | 72.9 | 38.4 | 61.9 |
| | Window-6 | **82.0** | 52.5 | 37.8 | 55.0 | 89.4 | 97.3 | 73.4 | 40.2 | 66.0 |
| | **Clip-12** | 81.0 | 54.5 | 39.8 | **81.8** | **92.0** | **98.9** | **75.5** | **45.3** | **71.1** |

## 3.3 MAIN RESULTS

To evaluate how our clip method affects trained models, we employ reject sampling to construct high-quality training datasets for each environment and trained a model named Qwen3-8B-RFT. Detailed training specifications are provided in Appendix D.3.

The results in Table 1 reveal several key insights: (1) **Clip Improvement**: Our clip context method achieves improvements over sliding window baselines with less hyperparameter dependency, with Qwen's best clip configuration reaching 68.9% compared to the best window baseline of 64.9%. (2) **Cross-Model Generalization**: The clip approach generalizes across different model architectures, with GPT-4o-mini demonstrating substantial gains (71.1% vs 66.0%) and Llama showing improvements from clip context (56.8% vs 55.3%). (3) **DPO Training Enhancement**: The DPO-trained model achieves 72.5% with clip context compared to the base model's 68.9%. (4) **LLM-based summarization does not outperform clipping**: When the summarizer uses the same size model, the Sum-12 approach is comparable to baseline Clip-12.

Different environments exhibit varying degrees of improvement. Environments that generate state transitions and produce new observations show larger improvements, with detailed environment discussion provided in Appendix O.

## 3.4 HYPERPARAMETER ANALYSIS OF CLIP CONTEXT

To understand the impact of clip context hyperparameters and provide guidance for practical application, we conduct comprehensive ablation studies varying both the clearing interval H and retention length L. Table 2 presents average performance across eight environments under different configurations. Several key insights emerge from this analysis:

Table 2: Ablation study of clip context hyperparameters. Performance (average success rate % across 8 environments) is shown for different combinations of clearing interval H and retention length L. Window context methods (marked with W) represent the special case where H=L+1. Empty cells indicate configurations not evaluated in this study.

| L \ H | H=2 | H=3 | H=6 | H=7 | H=12 | H=13 |
|---|---|---|---|---|---|---|
| L=1 | 62.2 (W=1) | 66.4 | 68.7 | - | **68.9** | - |
| L=6 | - | - | - | 64.9 (W=6) | 62.1 | - |
| L=12 | - | - | - | - | - | 61.3 (W=12) |

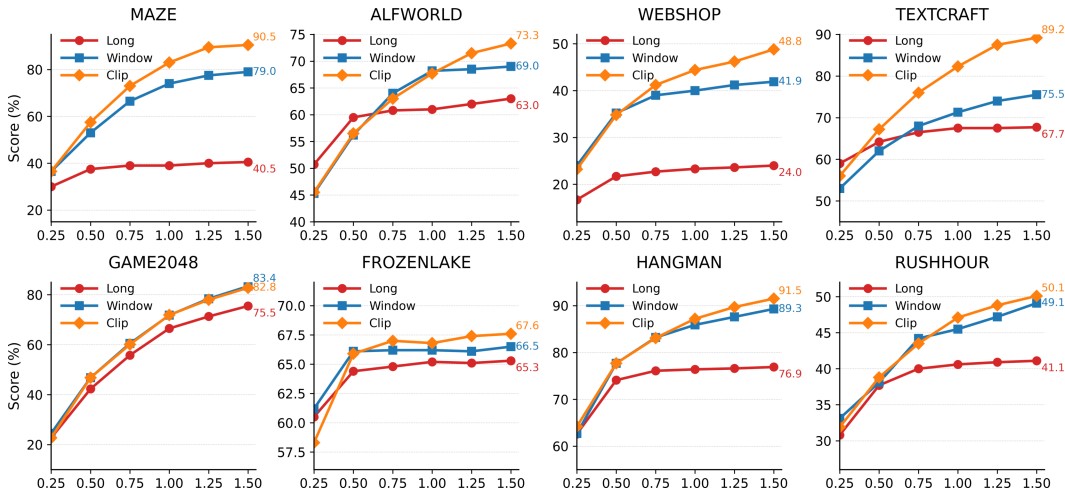

Figure 5: Score vs. maximum step scaling analysis. The x-axis represents the ratio relative to the baseline maximum step number.

**Effect of H parameter (clearing interval):** When L=1, increasing H from 2 to 12 shows consistent performance improvement. Larger H values allow the model to accumulate more contextual information for decision-making before clipping, leading to better exploitation of interaction history.

**Effect of L parameter (retention length):** When H=12, decreasing L from 12 to 1 shows consistent performance improvement. Lower L values provide stronger inertia breaking by retaining minimal history after clipping, which is effective for navigation environments. For long-term tasks as shown in Section 3.8, moderate L values may be preferable.

### 3.5 LONG-HORIZON SCALING ANALYSIS

To demonstrate the scalability advantages of our approach, we analyze performance across varying episode lengths from 0.25x to 1.5x standard evaluation limits, as shown in Figure 5. The scaling analysis reveals that our clip method outperforms window approaches in long episode settings by breaking inertia through periodic context clearing. The results also support our hypothesis that conversational inertia is one of the primary factors constraining long-horizon agent performance. The scaling behavior across environments further supports the generalizability of our approach for multi-turn interactions.

### 3.6 PRESERVATION OF GENERAL CAPABILITIES AFTER CONTEXT PREFERENCE LEARNING

We evaluate our Context Preference Learning trained models on standard benchmarks measuring general knowledge and reasoning capabilities in Table 3. The results demonstrate that Context Preference Learning preserves the model's original capabilities remarkably well.

Table 3: General capability preservation of DPO-trained models. GPQA-Diamond scores are averaged over 10 runs. We follow the same sampling parameter in Qwen3 (Yang et al., 2025).

| Mode | Model | GPQA-Diamond | MMLU-Redux |
|---|---|---|---|
| Non-thinking | Qwen3-8B | 48.83 | 79.13 |
| Non-thinking | Qwen3-8B-DPO | 49.09 | 79.32 |
| Thinking | Qwen3-8B | 58.03 | 83.04 |
| Thinking | Qwen3-8B-DPO | 57.92 | 83.27 |

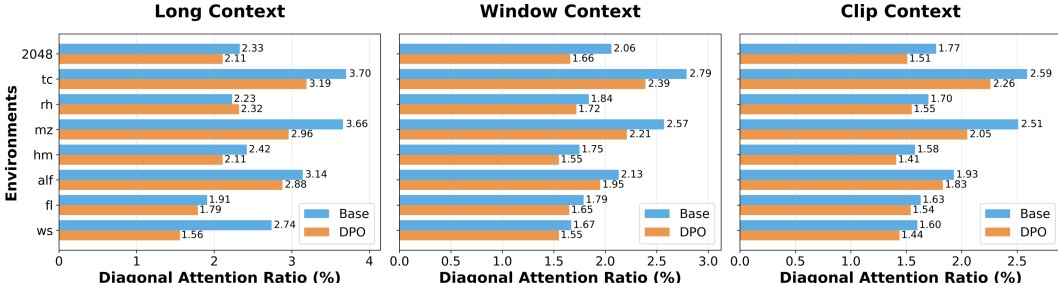

Figure 6: Diagonal Attention ratio analysis across different environments and context configurations. Lower ratios indicate reduced conversational inertia.

## 3.7 ATTENTION PATTERN ANALYSIS

To investigate how clip context methods and Context Preference Learning influence models at the attention level, ultimately enabling their decisions to break free from conversational inertia, we design attention-ratio indicators that provide quantitative insights into attention pattern changes.

We extract attention patterns by performing a single forward pass through the model and retrieving the attention matrix from the final layer. All attention heads are averaged to create a unified attention matrix. We categorize tokens based on their positional roles: sink tokens (first 3 tokens) , system tokens, user tokens, previous assistant tokens, and current assistant tokens. For each output token, we calculate the sum of attention weights directed toward each token category, then average these sums across all output tokens to obtain the final attention ratios for each category. Additionally, we introduce a diagonal-attention indicator that measures conversational inertia more precisely. This metric computes attention between the current output and previous assistant responses that occupy corresponding diagonal positions with an expansion of $r = 5$ tokens around the diagonal.

Through our analysis in Figure 6, we observe several findings: (1) **Clip context reduces diagonal attention**: Clip context exhibits lower diagonal attention compared to window methods, primarily because clipping to recent context enables greater exploration and reduces conversational inertia. (2) **DPO reduces diagonal attention stronger when baseline is higher**: DPO-trained models demonstrate lower diagonal attention than base models, attributed to the effectiveness of preference data in training the model to avoid imitation patterns. (3) **Targeted diagonal attention optimization**: DPO methods show greater variation in diagonal attention than in overall assistant attention, indicating that these approaches directly target and optimize the diagonal attention component that drives conversational inertia, while assistant attention ratios maintain a moderate range to balance exploration-exploitation. See Appendix A for additional experimental data and discussion.

## 3.8 COMPUTATIONAL EFFICIENCY ANALYSIS

To validate our theoretical KV cache optimization advantages, we conduct comprehensive speed benchmarking (with KV cache) across eight environments. As shown in Table 4, the experimental results validate our theoretical analysis and demonstrate significant computational advantages. **(1) Prefill Efficiency**: Our clip context method achieves the fastest prefill times across all environments, with 2-7x speedup over window context methods that cannot leverage KV cache due to continuously shifting boundaries. **(2) Stable Prefill Speedup**: While long context suffer speed

Table 4: Computational efficiency comparison across environments. Total inference time is the sum of prefill and generation time. Times reported in seconds.

| Timing | Context | MZ | WS | ALF | TC | HM | 2048 | FL | RH | Avg |
|--------|---------|------|------|------|------|------|------|------|------|------|
| Prefill | Full | 0.049 | 1.089 | 2.361 | 0.053 | 0.048 | 0.049 | 0.067 | 0.049 | 0.529 |
| Prefill | Window | 0.092 | 0.424 | 0.269 | 0.067 | 0.125 | 0.071 | 0.223 | 0.061 | 0.186 |
| Prefill | **Clip** | **0.048** | **0.057** | **0.052** | **0.044** | **0.047** | **0.038** | **0.061** | **0.047** | **0.055** |
| Total | Full | 2.934 | 4.560 | 4.776 | 2.609 | **3.770** | 2.878 | 3.102 | 5.323 | 4.212 |
| Total | Window | 3.222 | 3.610 | 2.205 | **2.534** | 3.781 | 3.607 | 2.617 | 3.187 | 3.483 |
| Total | **Clip** | **2.673** | **2.910** | **1.879** | 2.673 | 4.213 | **2.865** | **2.445** | **3.106** | **3.204** |

Table 5: Performance comparison on deep research task (BrowseComp (Wei et al., 2025) 128x2 case). Proactive answers occur when agents answer before reaching step limits, while forced answers occur at the step limit. Implementation details in Appendix J.

| Method | Score ($\pm$ SEM) | Proactive rate | Proactive acc. | Forced acc. |
|--------|-------------------|----------------|----------------|-------------|
| Window 6 | $25.0 \pm 1.7$ | 27.3% | 55.7% | 13.4% |
| Window 9 | $23.4 \pm 1.5$ | 26.6% | 58.8% | 10.6% |
| Window 12 | $24.2 \pm 1.8$ | 25.4% | 58.5% | 12.5% |
| Clip 12to0 | $25.0 \pm 1.8$ | 29.3% | 56.0% | 12.2% |
| Sum 12to0 (ReSum) | $27.7 \pm 2.0$ | 63.7% | 37.4% | 10.6% |
| **Clip 12to6** | **29.3** $\pm$ 1.5 | 30.9% | 61.0% | 15.1% |
| Sum 12to6 | $28.1 \pm 1.6$ | 30.8% | 63.3% | 13.2% |

degradation in complex environments which may exceed the context length. In contrast, our clip context method maintains minimal prefill overhead regardless of environment, confirming stable and more applicable performance under real-world conditions.

To evaluate the effectiveness of our clip context method in long-term reasoning scenarios that require deep research and multi-step thinking, we conduct experiments on BrowseComp. We compare our clip context method against window context and summarization-based approaches, specifically ReSum which uses LLM-generated summaries to manage context.

Table 5 presents results distinguishing between proactive answers (agents answer before reaching step limits) and forced answers (at maximum step). The results reveal three key insights: (1) **Model-generated summaries provide less accurate information than recent history**: Sum 12to0 (ReSum) underperforms Clip 12to6. (2) **Clip balances exploration and exploitation in long-term scenarios**: Window 9 underperforms Clip 12to6 despite having the same average input length. (3) **Additional summarization does not improve performance with sufficient historical examples**: Sum 12to6 achieves comparable performance to Clip 12to6.. More detailed analysis can be found in Appendix K

## 4 RELATED WORKS

Recent advances in large language models (LLMs) have made In-Context Learning a central topic in NLP. Brown et al. (2020) showed that GPT-3 can adapt to new tasks from a few examples, but later studies found its performance highly sensitive to prompt design (Lu et al., 2022; Webson & Pavlick, 2022; Sclar et al., 2024), indicating that ICL relies on surface-level pattern imitation rather than explicit semantic learning (Min et al., 2022). At the mechanistic level, induction heads in Transformers have been shown to implement copying behavior by attending to prefix patterns and replicating them, thereby offering a plausible basis for ICL (Olsson et al., 2022; D'Angelo et al., 2025). Along this direction, Halawi et al. (2024) identified two failure modes, namely overthinking and false induction heads, and proposed a prefix matching score to detect and suppress such heads. Nevertheless, these insights face challenges in long-context scenarios where LLMs exhibit positional bias (Mikhail et al., 2025; Hsieh et al., 2024b) and multi-turn performance degradation (Hong et al., 2025). Laban et al. (2025) found that premature answer attempts and verbosity cause degradation,

which can be consequences of conversational inertia, discussed in Appendix G. Related work has examined context effects in task-switching scenarios, where switching between different tasks within a conversation leads to performance degradation and task interference (Hankache et al., 2025; Gupta et al., 2024; Castillo-Bolado et al., 2024).

As LLM capabilities have expanded, agentic applications have emerged as a promising frontier. Most prominently, the ReAct framework (Yao et al., 2023b) integrates reasoning with action through alternating reasoning–action paradigms, while enhanced base agents improve decision-making via self-prompting and state-tracking (Rozanov & Rei, 2025). Complementary strategies introduce exploration–exploitation mechanisms through multi-path sampling (Wang et al., 2023) and systematic search approaches (Zhou et al., 2023; Yao et al., 2023a). Planning-based approaches have shown promise through explicit planning separation (Erdogan et al., 2025) and look-ahead strategies (Verma et al., 2025). Training methodologies have evolved with synthetic self-reflected trajectories (Chen et al., 2025) and context summarization for long-horizon tasks (Wu et al., 2025), though challenges persist with identity drift (Choi et al., 2025) and sophisticated reasoning requirements (Zheng et al., 2025; Zhang et al., 2025a;b). Multi-agent orchestrator-based systems have explored collaborative approaches through emergent behaviors (Chen et al., 2024), adaptive team building (Song et al., 2025), and optimizable graph structures (Zhuge et al., 2024), though these require environment-specific architectures and task decomposition capabilities. However, these approaches rely on complex pipeline designs that only indirectly alleviate conversational inertia. They do not address the fundamental nature of long-horizon agent problems and lack explicit mechanisms to tackle error propagation caused by induction heads.

## 5 DISCUSSION

### 5.1 CONTEXT CLIPPING AND INFORMATION RETENTION

Clipping retains moderate history, with navigation environments achieving optimal performance at minimal retention (Clip 12to1) and reasoning tasks benefiting from moderate retention (Clip 12to6). LLM-based summarization achieves comparable performance despite additional overhead, indicating compression benefits are offset by information loss. Clipping extends the principle that longer context does not uniformly improve performance—Window context outperforms Long context, and clipping amplifies this by removing larger chunks at once, more effectively mitigating inertia while exploiting strategic history management.

### 5.2 INFORMATION LOSS IN HISTORY DROPPING

Context clipping effectively mitigates conversational inertia but encounters limitations in ultra-long-horizon dependency tasks. For instance, in Test-Time Learning tasks (Hu et al., 2025) where early demonstrations fall outside the clipped window, failure necessarily occurs. This reflects a general unsolved challenge: window-based and summarization methods incur information loss, while long-context methods face fixed length constraints. Appendix L shows sequential tasks benefit from longer context while non-intuitive operation tasks favor shorter context's reduced inertia.

## 6 CONCLUSION

This work identified conversational inertia as a previously unrecognized factor limiting multi-turn agent performance. Our analysis reveals that diagonal attention patterns to previous responses emerge at moderate conversation lengths and constrain exploration capabilities. The proposed context bias calibration framework—combining clip context and Context Preference Learning—demonstrates effectiveness in mitigating these patterns. Experimental results across eight environments indicate that addressing conversational inertia can improve agent performance while reducing computational overhead, suggesting this phenomenon merits further investigation in multi-turn system design.

## 7 REPRODUCIBILITY STATEMENT

To ensure reproducibility of our results, we provide comprehensive implementation details throughout the paper and appendices. Section 3.2 specifies the exact models used (Qwen3-8B, Llama3.1-8B-Instruct, GPT-4o-mini), evaluation temperature (0.8), and hyperparameters for both clip context (H=12, L=1) and window context (W=6) methods. Our Context Preference Learning approach uses LoRA fine-tuning with detailed configurations in Appendix D.1, including rank=16, alpha=16, learning rate=5e-7, and dataset collection procedures with K=20 step minimum context gaps across 1,000 preference pairs per environment. Environment specifications are provided in Appendix N with exact step limits, reward structures, and evaluation protocols for all eight environments using the AgentGym framework. Appendix D.3 details reject sampling training configurations, while our prompt construction methodology with sequential message formatting and observation structuring is described in Appendix P. All attention analysis procedures are specified in Section 3.7 with r=5 token diagonal expansion and final-layer attention extraction methods.

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

## A   ATTENTION DISTRIBUTION ANALYSIS DETAILS

This section presents additional attention ratio data and discussion, as complementing Section 3.7.

As shown in Figure 2(c), different attention component ratios exhibit distinct trends as the number of context rounds increases. We hypothesize that diagonal attention in previous assistant responses primarily plays a harmful role in behavioral imitation. However, assistant attention, user attention, and system attention should be maintained within moderate ranges to balance exploration-exploitation. Although excluding any history from context can increase exploration by making system attention much stronger and user/assistant attention lower, this approach misses important contextual information needed to better exploit interaction history, resulting in significantly low performance as shown in Figure 2(b).

The experimental results in Figure 7 show two key findings: (1) Context Preference Learning significantly reduces diagonal attention while maintaining minimal changes to overall assistant attention. This successfully redirects diagonal attention to other parts of the assistant's processing, enhancing holistic understanding rather than positional imitation, while keeping user and system attention relatively stable. (2) Context clipping reduces attention to both assistant and diagonal components

while slightly increasing attention to system and user inputs. By directly controlling context, this approach focuses more on the balance of exploration-exploitation rather than learning from self-generated few-shot examples of unreliable quality.

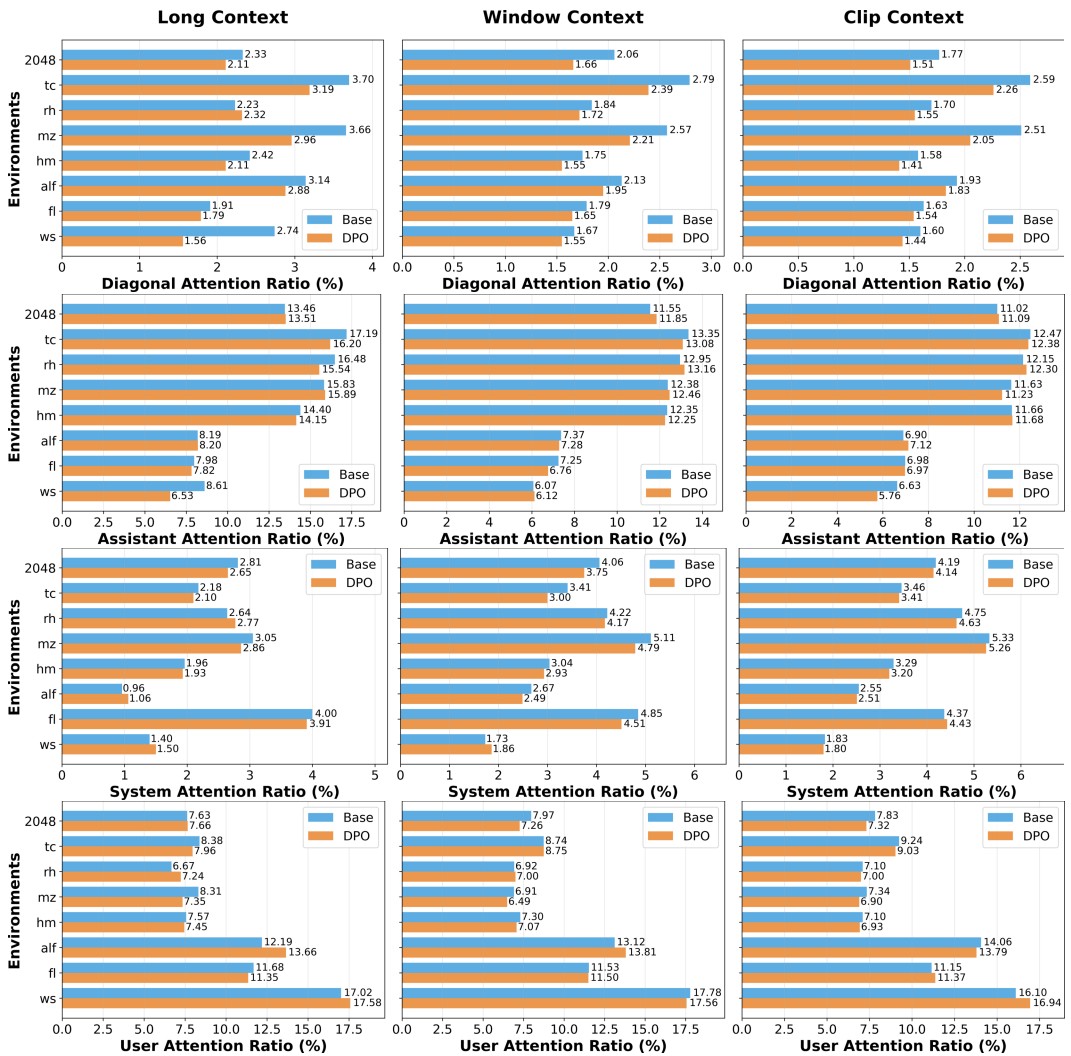

Figure 7: Attention distribution analysis across different component types and context management strategies. From top to bottom: diagonal attention (self-attention to previous assistant responses), assistant attention (attention to all assistant tokens), system attention (attention to system prompt tokens), and user attention (attention to user input tokens). Each subplot compares base models and DPO-trained models across three context management approaches: Long (full context), Window (recent context only), and Clip (filtered context).

## B    CONTEXT INFLUENCE AND BREAKING INERTIA ANALYSIS IN MAZE

To investigate how initial context influences subsequent model behavior and evaluate the effectiveness of different context management methods in breaking conversational inertia, we conduct a controlled experiment in the maze environment. As shown in Figure 8, we fix the starting state (position and task) while designing two different initial contexts. The two context types are: (1) a *good init example* containing an optimal trajectory (left, left, up, up) that moves directly toward the current position, and (2) a *bad init example* featuring a suboptimal looping pattern (right, left, right,

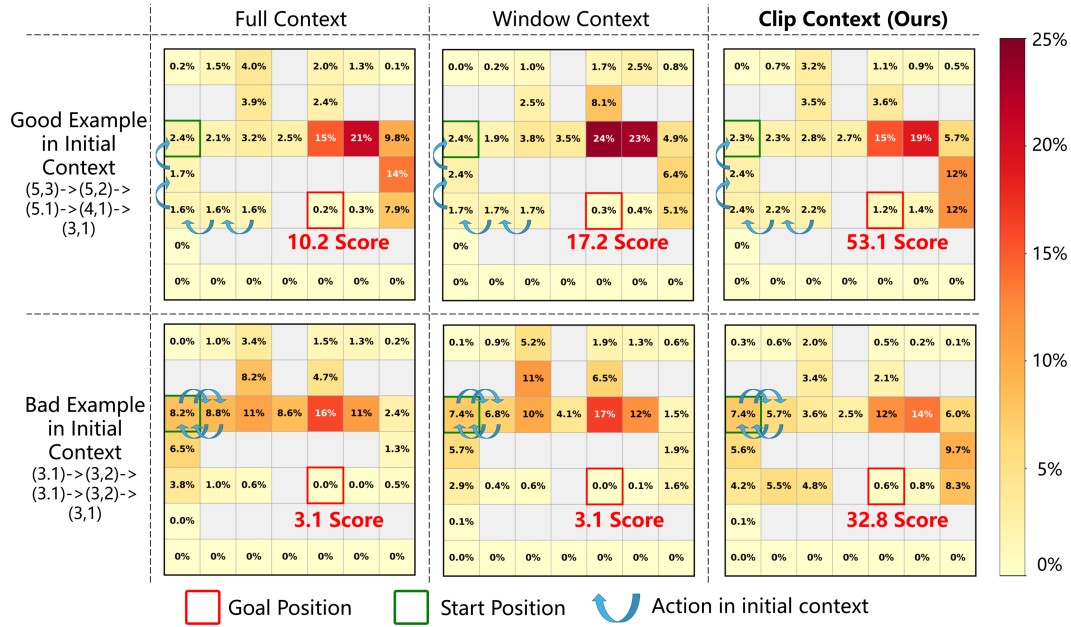

Figure 8: Context influence analysis in maze environment. We compare three context management methods (Full context, Window context, and Clip context) under both good and bad initial contexts. Each heatmap shows visit frequency as percentages (e.g., 0.3% means the agent spent 0.3% of total steps at that position). Scores in the tables indicate overall task performance.

left) that wastes steps. This design aims to verify whether agents follow initial context examples and whether inertia phenomena exist that lead to performance differences.

Our experimental findings reveal two key insights that validate the existence of conversational inertia and demonstrate our method's effectiveness. (1) **Strong Context Following Behavior**: Models exhibit pronounced sensitivity to initial context, with bad initial examples leading to suboptimal exploration patterns across all methods. This empirically confirms that conversational inertia significantly impacts agent behavior in multi-turn scenarios shown in Figure 1. (2) **Clip Context Effectiveness**: Our clip context method successfully mitigates the negative impact of poor initial context while maintaining effective exploration strategies. Unlike full context and window context methods that remain heavily influenced by suboptimal initial examples, clip context demonstrates robust performance regardless of initial context quality.

We fix the starting state (position and task) while designing two different initial contexts. The two context types are: (1) a *good init example* containing an optimal trajectory (left, left, up, up) that moves directly toward the current position, and (2) a *bad init example* featuring a suboptimal looping pattern (right, left, right, left) that wastes steps. We execute 64 episodes with a 60-step limit from identical starting states. We visualize position visit frequencies as heatmaps to analyze exploration patterns and step efficiency across context management methods, where visit percentage represents the proportion of total steps spent at each position. Window context uses $W = 6$, while Clip context maintains $H = 12$, $L = 1$.

As shown in Figure 8, our experimental findings reveal several key insights:

**Strong Context Following Behavior:** Comparing the first row (good init example) with the second row (bad init example) across all three columns shows dramatic performance differences. We control for identical starting states, action spaces, and optimal paths. However, performance varies drastically between good and bad init examples. This demonstrates that models incorrectly transfer strong few-shot learning capabilities, automatically treating previous agent actions in context as examples to imitate, regardless of their quality. This finding aligns with our observations in Figure 1,

confirming that models can significantly degrade performance when exposed to patterns that are implicitly accepted without scrutiny.

**Persistent Context Influence and Window Context Limitations:** Examining the Window context column (middle column) reveals that the quality of the initial 4-step context profoundly affects subsequent 60-step performance across all context management methods. Even though early history is quickly dropped in both Window context and Clip context approaches, the influence of good versus bad init examples persists throughout episodes, significantly impacting final scores. This demonstrates the lasting effect of early context on model behavior and shows that simply limiting recent history cannot break error loops. While Window context provides modest improvements under good init examples compared to Full context, it fails to improve performance under bad init examples, highlighting its inability to escape negative patterns. Although Window context removes outdated steps and excludes the initial 4-step bad example, it consistently maintains the most recent 6-step context, causing negative influences to persist throughout episodes and resulting in extremely low completion rates.

**Clip Context Effectiveness:** Comparing the third column (Clip context) with the first two columns shows that only Clip context successfully mitigates the impact of bad init examples. Most notably, Clip context achieves 32.8 score under bad init examples compared to Window context's 3.1, demonstrating a 10.6× improvement in breaking free from negative patterns. Analysis of visit percentage distributions reveals that Clip context shows fewer instances of getting stuck at individual positions compared to Full context or Window context methods. In the optimal direction (5-th row, rightmost position), Clip context achieves 12% visit frequency under good init examples and 8.3% under bad init examples, higher than Window context's 5.1% and 1.6% respectively. Conversely, in suboptimal directions (upper right regions moving away from the goal), Clip context shows lower visit frequencies than Window context. Furthermore, under good init examples, Clip context's highest single-position visit percentage is 19%, lower than Full context's 21% and Window context's 24%, indicating more balanced strategic exploration.

## C SUPPLEMENTARY EXPERIMENTAL RESULTS

This section provides additional experimental results and detailed ablation studies that support the findings presented in the main paper.

### C.1 DETAILED HYPERPARAMETER ABLATION STUDIES

We provide detailed ablation studies examining the individual effects of H and L parameters, as well as comprehensive comparisons between clip and window methods at various context lengths.

**Effect of H Parameter:** Table 6 shows how varying the clearing interval H affects performance when L=1. The results demonstrate consistent improvement as H increases from 2 to 12, with diminishing returns beyond H=12.

Table 6: Effect of H parameter on performance (L=1 fixed). Scores represent average success rates across 8 environments.

| Configuration | L=1, H=2 | L=1, H=3 | L=1, H=6 | L=1, H=12 |
|---|---|---|---|---|
| 8 env avg Score | 62.2 | 66.4 | 68.7 | 68.9 |

**Effect of L Parameter:** Table 7 examines how the retention length L impacts performance. Lower L values provide stronger inertia breaking, leading to better performance in the evaluated environments.

Table 7: Effect of L parameter on performance. Window 12 is equivalent to Clip with L=12, H=13.

| Configuration | L=1, H=12 | L=6, H=12 | L=12, H=13 (Window 12) |
|---|---|---|---|
| 8 env avg Score | 68.9 | 62.1 | 61.3 |

**Comprehensive Clip vs Window Comparison:** Table 8 presents a systematic comparison between clip and window methods across multiple context length configurations. The results demonstrate that clip consistently outperforms window methods when the average input length is controlled, with the performance gap widening as context length increases.

Table 8: Systematic comparison between clip and window methods at equivalent average input lengths. Clip methods consistently outperform window approaches across various context length settings.

| Configuration | Clip (L=1, H=?) | Window (W=?) |
|---|---|---|
| L+H=3, W=1 | 62.2 | 62.2 |
| L+H=7, W=3 | 68.7 | 64.0 |
| L+H=11, W=5 | 68.8 | 64.2 |
| L+H=13, W=6 | 68.9 | 64.9 |
| L+H=15, W=7 | 68.9 | 63.8 |

### C.2 ABLATION STUDY ON PREFERENCE PAIR CONSTRUCTION

To validate the effectiveness of our Context Preference Learning approach, we conduct ablation studies on different preference pair construction strategies. Table 9 compares three configurations: baseline without DPO training, preferring short (1 round) over moderate (6 rounds) contexts, and preferring moderate (6 rounds) over excessively long contexts.

Table 9: Ablation study on DPO preference pair construction (Qwen3-8B base, Clip 12to1 evaluation). Results show that preferring moderate-length contexts over excessively long contexts achieves the best performance.

| Chosen | Rejected | 8 env avg score |
|---|---|---|
| - | - | 68.9 (Base) |
| 1 round | 6 rounds | 70.3 |
| 6 rounds | As long as possible | 72.5 |

The results reveal that both preference construction strategies improve performance over the baseline. However, preferring moderate contexts over excessively long contexts (6 vs long) achieves superior results (72.5) compared to preferring short over moderate contexts (1 vs 6, score 70.3). This finding validates our hypothesis that the most effective approach is to mitigate the negative effects of excessive inertia in overly long contexts, rather than simply maximizing exploration through minimal context.

## D ADDITIONAL TRAINING EXPERIMENTAL DETAILS

### D.1 TRAINING CONFIGURATION

We implement DPO training using LoRA (Low-Rank Adaptation) fine-tuning, which approximates weight updates through low-rank matrices to achieve parameter-efficient training. For a weight matrix $W \in \mathbb{R}^{d \times k}$, LoRA decomposes the update as:

$$W' = W + \frac{\alpha}{r} \cdot A \cdot B \tag{3}$$

where $A \in \mathbb{R}^{d \times r}$ and $B \in \mathbb{R}^{r \times k}$ are trainable low-rank matrices. The rank $r$ determines the dimensionality of the low-rank decomposition (controlling model expressiveness), while alpha $\alpha$ controls the scaling factor for the adapted weights (regulating adaptation strength). Our main training configuration employs LoRA fine-tuning with rank 16 and alpha 16, targeting 0.4% of model parameters. We use DPO loss with beta=0.01, and sigmoid loss type. Training runs for 2 epochs with learning rate 5e-7 using AdamW optimizer.

Our DPO data collection uses a minimum context gap of $K = 20$ steps to ensure sufficient disparity between long and short context windows. We collected 1,000 preference pairs per environment, yielding an 8K total dataset across all environments.

Hardware setup includes 4×A100 GPUs for training and attention visualizing, 8×RTX4090 for evaluation, and 1×A6000 for computational efficiency benchmarking.

## D.2 LoRA Ablation Study

To investigate whether the model learns environment-specific patterns or develops generalizable capabilities, we conduct DPO training using an Ultra-Low Parameter (Rank=1) experiment. By severely constraining the trainable parameter count to be far smaller than the dataset size, this setup prevents environment overfitting and forces the model to learn unified capabilities across environments. We use the same eight environment dataset as the main experiments but with extremely constrained parameters. We employ LoRA rank=1 with alpha=32, resulting in a high alpha/rank ratio of 32. To ensure training stability with this aggressive parameter reduction, we implement gradient accumulation steps of 4 and DPO label smoothing of 0.1. This configuration trains only 0.02% of the model parameters while maintaining effective learning dynamics. All other training parameters remain identical to the main DPO configuration.

Besides, to understand the impact of hyperparameter choices on our reward-free DPO training approach, we conduct ablation studies on LoRA alpha values while maintaining rank=16. We evaluate three configurations: alpha=12, alpha=16, and alpha=24 across all eight environments using Qwen3-8B as the base model.

Table 10: Ablation study of LoRA training parameters for DPO. Success rates (%) are reported for different parameters. Evaluation with Clip Context Strategy

| Rank | Alpha | MZ | ALF | WS | TC | FL | HM | 2048 | RH | Avg |
|------|-------|------|------|------|------|------|------|------|------|------|
| 1 | 16 | 89.0 | 67.0 | 48.7 | **86.0** | **71.6** | 85.6 | 72.4 | 48.0 | 71.0 |
| 16 | 12 | 86.0 | **73.0** | 50.7 | 84.0 | 69.4 | 83.6 | **74.0** | 49.1 | 71.2 |
| 16 | 16 | **91.5** | 70.3 | **54.9** | 83.0 | 68.4 | 87.5 | 73.0 | 51.1 | **72.5** |
| 16 | 24 | 91.0 | 70.5 | 51.2 | 79.5 | 67.2 | **88.8** | 72.9 | **52.3** | 71.7 |

The ablation results reveal that alpha=16 provides the optimal balance across most environments, achieving the highest average performance of 72.3%. While alpha=24 shows competitive performance in certain environments (ALF, HM, RH), it exhibits reduced performance in strategic reasoning tasks (TC, WS, 2048). This suggests that moderate regularization strength (alpha=16) better preserves the model's original capabilities while effectively incorporating the conversational inertia mitigation preferences.

## D.3 Reject Sampling Training Details

For evaluating our clip context method's effectiveness on supervised fine-tuned models, we constructed high-quality training datasets using reject sampling using Qwen3-8B base model. Each environment contributed approximately 1000 successful episodes, with sampling conducted under identical step limits used in evaluation protocols.

Table 11: Training dataset composition across environments. Data collected through reject sampling with perfect episode retention.

| Environment | RH | MZ | WS | ALF | TC | FL | HM | 2048 | Total |
|-------------|------|------|------|------|------|------|------|------|-------|
| **Steps** | 497 | 1013 | 991 | 1069 | 1178 | 991 | 484 | 1120 | 7343 |

The training configuration employed 3 epochs with a learning rate of 2e-5, a linear warmup ratio of 0.03, and cosine learning rate scheduling. The hardware setup utilized 4×A100 GPUs with gradient accumulation steps of 8 for stable training dynamics.

## E Continuous Cache Pruning vs Discrete History Truncation

This section first describes and categorizes the work on KV Cache optimization, then explains why Window Context cannot use KV Cache Reuse.

A large body of work on attention optimization for LLMs can be broadly grouped into two paradigms: continuous pruning and discrete pruning. The distinction lies not in the granularity of tokens versus rounds, but in whether pruning is performed in a continuous or blockwise fashion.

Continuous pruning methods operate in a fine-grained, token-continuous manner. Representative approaches include sliding-window attention, KV-cache eviction, and other forms of local or structured sparsity. These methods prune context progressively, allowing partial information flow to be preserved across adjacent segments. Their theoretical complexity is low and they are widely used in long-context modeling.

Discrete pruning methods operate in a coarse-grained, blockwise manner. Examples include prefix caching, history clearing, and round-level memory selection in multi-turn agent settings. Instead of preserving continuity, these methods explicitly remove or reset entire chunks of context, which often aligns better with the online nature of interactive agents.

Remark on the Window Context method. Although the sliding-window attention mechanism is often discussed under long-context modeling, in the present work we treat it as a discrete-level pruning strategy as nearly all agent works do. This is because each window defines a hard cutoff beyond which past information cannot be reused, preventing effective KV-cache reuse across multiple dialogue turns. If we forcibly apply KV cache reuse in Window Context, the results would theoretically differ from the non-optimized Window Context version, effectively becoming an approximation of Full Context with stronger inertia effects. As a result, Window Context cannot utilze KV Cache between rounds theoretically and is generally inefficient for agent-style tasks.

## F  COMPUTATIONAL COMPLEXITY ANALYSIS

We analyze the computational advantages of our approach through prefill cost comparison:

*Window Context Complexity:* With window size $W$, each inference step processes a context of length $W$ rounds. Since the oldest round is dropped at each step, prefix caching cannot be utilized due to constantly changing context boundaries. The prefill cost per step is $O(W^2)$ attention operations.

*Clip Context Complexity:* Our method alternates between context lengths $L, L+1, \ldots, H-1$ within each clearing cycle. Crucially, all contexts within a cycle share the same prefix up to the last clearing point, enabling effective KV cache utilization. Specifically, when processing context length $i+1$, the KV cache from the previous context length $i$ can be reused, requiring only incremental computation for the new round. This means only the marginal cost $O(i)$ (instead of $O(i^2)$) is needed for each step after the initial prefill. The average prefill cost becomes:

$$\text{Average Cost} = \frac{O(L^2) + \sum_{i=L+1}^{H-1} O(i)}{H - L} = \frac{O(L^2) + O((H-1)^2 - L^2)/2}{H - L} \tag{4}$$

To compare with sliding window methods fairly, we consider the case when $L = 1$ and $H = 2W$ (equivalent total context budget, maintaining the same average input rounds for fair computational comparison). The speedup ratio becomes:

$$\text{Speedup} = \frac{\text{Window Cost}}{\text{Clip Cost}} = \frac{W^2}{\frac{1+((2W-1)^2-1)/2}{2W-1}} = \frac{2W^2(2W-1)}{2 + (2W-1)^2} \approx W \tag{5}$$

This analysis demonstrates that our clip context method achieves approximately $W \times$ speedup in prefill computation compared to sliding window approaches, with the computational advantage stemming from KV cache reuse within each clearing cycle.

## G  DIAGONAL ATTENTION: ALTERNATIVE EXPLANATIONS AND CAUSAL DIRECTION

Prior mechanistic work on induction heads and prefix copying (D'Angelo et al., 2025) provides theoretical support for our findings. These studies demonstrate that induction heads implement copying behavior by attending to previous patterns and replicating them, which aligns with our observed diagonal attention patterns. This mechanistic understanding suggests that diagonal alignment encodes

imitation dynamics that accumulate errors across turns, making it a plausible causal mechanism for *conversational inertia* rather than merely a correlational artifact.

Alternative explanations, such as premature answering and verbosity (Laban et al., 2025), can be interpreted as downstream symptoms of this inertia. Our Context Preference Learning selectively suppresses diagonal attention while leaving other attention flows intact, and this targeted reduction consistently improves performance (Figure 6). This selective intervention provides direct evidence that diagonal attention itself is harmful, supporting its role as a causal contributor rather than an incidental correlate.

## H  STATISTICAL SIGNIFICANCE AND VARIANCE ANALYSIS

To ensure the reliability of our experimental results, we report standard error of the mean (SEM) across all experiments. Table 12 presents SEM values for all model configurations across eight evaluation environments. The consistently low SEM values validate the statistical significance of our findings.

Table 12: Standard error of the mean (SEM) for all experimental configurations across eight environments. All values are reported as percentages. The low SEM values relative to performance differences validate the statistical significance of our findings.

| Model | Context | MZ | ALF | WS | TC | FL | HM | 2048 | RH | Avg |
|---|---|---|---|---|---|---|---|---|---|---|
| Qwen3-8B | Window-6 | 3.59 | 1.06 | 0.08 | 0.75 | 1.55 | 1.10 | 1.66 | 3.04 | 0.57 |
| Qwen3-8B | Clip-12 | 1.86 | 1.24 | 1.73 | 0.65 | 2.10 | 1.14 | 1.39 | 2.45 | 0.56 |
| Qwen3-8B-DPO | Clip-12 | 2.17 | 0.18 | 0.99 | 0.94 | 1.78 | 0.94 | 0.73 | 1.61 | 0.41 |

The uncertainty estimation methodology is as follows: for each task in each dataset, we sample multiple times to obtain multiple average scores per benchmark. We then calculate the standard error across these average scores for each benchmark separately. The overall aggregated uncertainty across all eight environments is computed as the sum of individual standard errors divided by $\sqrt{8}$, providing a normalized measure of cross-benchmark uncertainty.

## I  SUMMARIZATION IMPLEMENTATION DETAILS

We follow the prompting approach from ReSum (Wu et al., 2025). For each clip, we pass all context history along with the last summary (if available) to generate an updated summary that captures the essential information for decision-making.

## J  BROWSECOMP EVALUATION DETAILS

We evaluate clip context and summarization methods on BrowseComp (Wei et al., 2025). We use Tongyi Deep Researcher as the actor and Qwen3-30B-A3B-Instruct-2507 as the summarizer. We deliberately choose models of comparable size to ensure the summarizer does not leak information to the actor, which would allow the summary to substitute for decisions that should be made by the actor itself.

To reduce evaluation variance, we first run the same actor for 12 steps across all configurations. Subsequently, all settings continue from this common 12-step baseline, ensuring identical early-stage trajectories and reducing bias. We evaluate on the first 128 cases from BrowseComp, with each configuration run twice.

**Standard Error Calculation:** We report a standard error of the mean (SEM) that reflects **only within-case variability**, avoiding inflation caused by differences in case difficulty. For each evaluation case, we compute its empirical accuracy $\hat{p}_i$ from $k_i$ repeated judgments, and estimate its variance as $\hat{p}_i(1 - \hat{p}_i)/k_i$. Treating cases as fixed and randomness as arising solely from repeated

evaluations, the SEM of the overall accuracy is then computed as

$$\text{SEM} = \frac{1}{N}\sqrt{\sum_{i=1}^{N}\frac{\hat{p}_i(1-\hat{p}_i)}{k_i}},$$

where $N$ is the number of cases. This measures uncertainty due to evaluation noise rather than case difficulty.

**Evaluation Metrics:** We measure three complementary metrics: (1) Overall score based on answer correctness, (2) Proactive answer rate measuring the percentage of cases where agents answer before reaching step limits, indicating confidence in gathered information, and (3) Accuracy split between proactive and forced answers, revealing whether agents can accurately judge when sufficient information has been collected versus being forced to guess at the deadline.

**Summarization Implementation:** We implement the ReSum (Wu et al., 2025) summarization approach using their open-source codebase with minimal modifications to ensure compatibility with our experimental framework.

## K    EMPIRICAL STUDY OF SUMMARY-BASED CONTEXT MANAGEMENT

Existing summarization approaches show unstable performance, suffering from context collapse and brevity bias. Through our empirical study, we also found critical issues with summary-based methods, including over-confidence: model-generated summaries may lead the actor model to have higher proactive answer rates but lower answer accuracy. Therefore, our method does not adopt summarization. In contrast, our Clip method is simple yet effective, and we recommend it as a strong baseline.

### K.1    QUANTITATIVE ANALYSIS AND CASE STUDY

Our experiments reveal that from a quantitative perspective, summaries encourage the actor model to make decisions even when the prompt does not explicitly require it, while simultaneously reducing decision accuracy. This phenomenon becomes more pronounced when only summary content is retained after clipping, without bare history turn information, as shown in Table 5.

We conducted an empirical case study using Qwen3-30B-A3B-Instruct-2507 as the summary model. From a dataset of 15 failed actor cases, we analyzed whether the summaries contributed to the failures. We identified three major categories comprising four sub-categories of common errors:

**Over-claiming:** Making definitive assertions about ambiguous information.

- Using definitive language for uncertain matters, as demonstrated in Example 1.

**Premature Conclusion:** Drawing conclusions that should be deferred to the actor.

- Making absolute claims based on insufficient evidence, preempting the actor's judgment and ignoring exploration possibilities, illustrated in Example 2.

**Missing Information:** Omitting critical context or findings.

- Omitting critical failed path information, which may lead the actor to repeat verified error directions, exemplified in Example 3.
- Omitting previous actor key findings, which may cause the actor to redundantly search again.

Table 13 presents the distribution of error types as percentages.

Table 13: Distribution of Summary Error Types in Failed Cases

| Error Type | Percentage |
|---|---|
| Definitive language for uncertain matters | 66.7% |
| Overly absolute claims | 60.0% |
| Missing critical failed path information | 46.7% |
| Unmentioned key findings | 46.7% |

### K.2 REPRESENTATIVE EXAMPLES

We present representative examples demonstrating each error category with their problematic summaries highlighted.

---

**Example 1: Over-claiming** (Sample 22)

*Summary excerpt:* " The 1990s short-lived TV series that matches the criteria is 'Savannah,' which aired on ABC from August 23 to September 13, 1996. The actor born in Tennessee is Leslie Jordan, who was born in Memphis, Tennessee, and appeared in the series. The actor who was a Caribbean immigrant is Eva LaRue, who is of Mexican descent; however, no evidence in the conversation confirms she is a Caribbean immigrant . The conversation does not identify a Caribbean immigrant actor in the cast of 'Savannah.'"

*Issue:* The summary opens with a definitive assertion "The 1990s short-lived TV series that matches the criteria is 'Savannah,'" yet immediately contradicts itself by stating that no evidence confirms key criteria. This over-claiming problem uses absolute language like "matches the criteria" while the actual conversation failed to verify all three actor requirements, particularly the Caribbean immigrant and the law enforcement father criteria

---

**Example 2: Premature Conclusion - Absolute Claims** (Sample 5)

*Summary excerpt:* " The repeated mention of 'Bath UK author' across various search queries, despite the lack of verifiable content on the cited websites, indicates that Bath is the city associated with the author in the context of the alliterative titled short story published in September 2023."

*Issue:* This represents an unjustified leap from unreliable mentions to a definitive conclusion, preempting the actor's judgment. The summary acknowledges "lack of verifiable content" yet still "indicates" a definite answer, replacing the actor's decision-making role with a premature conclusion based on insufficient evidence.

---

**Example 3: Missing Information - Failed Path** (Sample 43)

*Summary excerpt:* "The conversation history does not provide sufficient information to confirm that Yuri Seung is the specific researcher in question or to determine the number of relevant PubMed publications."

*Issue:* While this acknowledges uncertainty, it fails to explicitly highlight that no researcher identity could be confirmed, which is critical information for the actor's decision-making. The summary omits the failed exploration paths that would help the actor understand why verification was impossible, potentially causing redundant searches.

> **Example 4: Maze GPT-5-mini - Bad orgnized** (Sample 7)
> *Summary excerpt:* "- Essential Information:
> - Goal position: (8, 6).
> - Current position progression:
> - Started at (1, 9) with a wall above.
> - Moved to (1, 8) with walls above and below.
> - At (1, 7), walls on left, above, and below; only right was open, leading back to (1, 8).
> - Repeated oscillations between (1,9) (1,8) and occasionally (1,7), indicating a vertical corridor
> at x=1 with vertical movement blocked at y=8,9,10 and left blocked at (1,7).
> - A single right move from (1,9) reached (1,10) where walls were above and below.
> - Last known position: (1, 9) with a wall above.
> - Constraints observed:
> - At (1,8) and (1,10): walls above and below (vertical movement blocked).
> - At (1,7): walls on left, above, below (only right allowed).
> - At (1,9): wall above; no info about other directions there.
> - Navigation outcome so far: No net progress toward goal; trapped cycling near x=1 with vertical
> blocked; exploration to the right from (1,8)/(1,9) appears necessary to escape."

## L    DISCUSSION ON INFORMATION LOSS

To empirically investigate the trade-off between information gain and inertia, we randomly selected 50 tasks in AlfWorld and evaluated them using the Qwen3-8B base model. Each task was executed 16 independent times under two configurations: Clip-12 and Clip-6. Clip-12 represents longer context retention but potentially stronger inertia, while Clip-6 provides less historical information but may enable more adaptive behavior.

We observed substantial performance gaps between the two settings across different task types. Notably, 20% of tasks exhibited success rate differences of 30% or more. Table 14 shows the distribution of success rate differences (Clip-12 minus Clip-6) across tasks.

Table 14: Distribution of success rate differences between Clip-12 and Clip-6 configurations in AlfWorld. Positive values indicate Clip-12 performs better; negative values indicate Clip-6 performs better.

| Difference Range | [-0.4, -0.3) | [-0.3, -0.2) | [-0.2, -0.1) | [-0.1, 0.1) | [0.1, 0.2) | [0.2, 0.3) | [0.3, 0.7) |
|---|---|---|---|---|---|---|---|
| Count | 1 | 1 | 3 | 24 | 10 | 2 | 9 |

Through qualitative analysis of tasks where each configuration excelled, we identified characteristic patterns consistent with the information-inertia trade-off:

**Clip-12 advantages:** In case 2541, which requires accessing multiple different interaction types, the sequential inspection task requires a memory of previously checked items and current state to avoid repeated exploration. Similarly, case 2582 involves coherent multi-step sequential operations with strong sequential dependencies. When context length falls below the dependency span, task success becomes nearly impossible. Clip-12's extended context enables proper tracking of these long-term dependencies.

**Clip-6 advantages:** In case 2531, the agent encounters ambiguous feedback such as "nothing happens," which typically signals operation failure in most environments. However, in AlfWorld, this feedback indicates successful execution without visible effects. A similar phenomenon is observed in case 2425, where the agent using Clip-12 encounters an unexpected result and believes it is unsolvable, so it repeatedly outputs an "exit" action until the history is clipped. Based on these case analyses, Clip-12 tends to persist with conventional interpretations due to stronger historical influence, ultimately failing to resolve the task. Clip-6, with less historical context and more frequent clipping, exhibits more flexible interpretation patterns and successfully adapts to the environment's unconventional feedback semantics.

This experiment demonstrates that information loss and inertia represent a fundamental trade-off, directly limiting the scalability of fixed-limit window-based context management approaches. For contexts within our maximum context round size, our clip method addresses this limitation by maintaining low inertia while providing higher information retention capacity, achieving a more favorable balance. However, tasks requiring context beyond the maximum context round size remain inherently challenging for our approaches.

Despite the above failure cases, we note that compared to Long context (which retains full history without any information loss), Clip-12to1 achieves a higher overall success rate on ALFWorld (Clip-12to1 achieves 67.7 while Long context achieves 61.0, as shown in Table 1). Additionally, we follow the same case study setting mentioned above and find that in tasks (e.g., Case 2531, 2425) with difficult non-intuitive operations, shorter context methods have less inertia and are less prone to being trapped in loops, thus having a higher probability of exploring correct directions. This demonstrates that while individual tasks may fail due to information loss, the overall performance benefits from reduced inertia.

**Information loss is an unsolved challenge across all context management approaches.** We emphasize that the information loss challenge is inherent to existing context management strategies, not unique to Clip. The Window method drops conversation turns beyond the window boundary. The summarization approach suffers from lossy and biased compression. As shown in Appendix K, summaries frequently exhibit over-claiming, premature conclusions, and missing critical information. Neither window methods nor summarization approaches adequately address the information loss problem. Our Clip method does not aim to address the information loss challenge. Instead, we identify and address a different problem—conversational inertia—which has been overlooked in prior work.

**Inertia reduction and information preservation can be addressed separately.** We agree that in some cases more information is valuable. However, our key insight is that inertia and information access are two different dimensions that can be addressed separately and combined effectively. Table 15 demonstrates this on BrowseComp. We can see that: (1) reducing inertia can outweigh information loss, as Clip-12to0 achieves better performance than Window-12 despite introducing some information loss by discarding history; (2) the information loss drawback can be recovered by combining Clip with summarization methods, where Sum-12to0 performs better than Clip-12to0; (3) alternatively, directly retaining some recent history as context also proves effective, as demonstrated by Clip-12to6 performing better than Clip-12to0.

Table 15: Combining Clip (reduce inertia) with information-preserving methods in BrowseComp

| Method | Score ($\pm$ SEM) |
|---|---|
| Window-12 | $24.2 \pm 1.8$ |
| Clip-12to0 | $25.0 \pm 1.8$ |
| Sum-12to0 | $27.7 \pm 2.0$ |
| Clip-12to6 | $29.3 \pm 1.5$ |

The above results imply that Clip's inertia reduction can be effectively combined with other information-preserving methods. Besides, the need to reduce inertia exists broadly in long-horizon multi-round agent scenarios: even a RAG-enhanced agent with task-relevant information retrieval still encounters conversational inertia problems when queries form multi-round interactions in its conversation context history. As a promising future direction, this inertia and information loss tradeoff could be mitigated by combining inertia-aware adaptive dropping with information-based retention, allowing critical earlier turns to be preserved while still reducing inertia.

## M   HOW TO SELECT CLIP HYPERPARAMETERS

Clip parameters can be easily selected based on task characteristics, with lower sensitivity compared to Window methods. As shown in General Response 3, Window context is a special case of

Clip (Window W is completely equivalent to Clip L=W, H=W+1). When we decrease L while increasing H, the average input information remains constant, but the model's disruption of inertia becomes stronger when refreshing context.

In practice, the H parameter of Clip can be kept at Window W or slightly higher. For the L parameter: in multi-turn scenarios where agent actions cause state transitions (e.g., navigation tasks), setting L to a lower value is sufficient (L=1, L=3). For tasks where each step involves equal reasoning (deep research or other multi-step reasoning), L needs to be set to a moderate value (L=6).

Adding a new hyperparameter L does not increase the difficulty of practical application. Window context is widely used in agent systems, and existing approaches (e.g., UI-TARS (Qin et al., 2025)) already require hyperparameter tuning based on experience. We believe Clip's dual parameters do not impose greater tuning burden than Window methods. According to the experimental tables, a moderate Clip parameter is sufficient to outperform carefully selected Window parameters.

## N   ENVIRONMENT SPECIFICATIONS

Different environments have different observation lengths. For full history implementation, we aim to input as many rounds as possible to agents. Therefore, we set the default full history window size to adapt to observation length.

**WebShop(WS)**: WebShop (Yao et al., 2022) is a simulated e-commerce platform where agents execute product procurement tasks adhering to predefined criteria through interface interactions. The environment integrates 12,000 structured instructions and leverages over one million real-world Amazon product listings, with 6,910 instructions selected for task execution. Agents can navigate through button-based interactions or utilize text-based search functionality to locate and purchase products meeting specific requirements. Performance is quantified via average score, with task sequences limited to 40 rounds to balance efficiency and practical applicability. When configured with full history strategy, we set the maximum history window to 35. We utilize the AgentGym library while maintaining the original system prompt, and format available actions as attachments to environment observations to provide comprehensive action space information.

For WebShop evaluation using GPT-4o-mini, we found that some product descriptions containing sensitive content caused evaluation model refusals. To ensure fair comparison, we controlled both methods to use nearly identical feasible evaluation samples.

**Maze(MZ)**: Maze is a grid-based navigation task from LMRL Gym (Abdulhai et al., 2023) where agents must reach fixed goal locations through strategic movement. At each step, agents move one cell in four directions (up, down, left, right), with observations indicating current position, goal position, and adjacent wall information. The environment supports both fully observable and partially observable variants, with the latter providing only action history. Evaluation employs binary success metrics: episodes score 1 if agents reach goals within step budgets and 0 otherwise. Maximum episode length is set to 60 steps, with identical limits for full history configurations. The deterministic nature of movement mechanics ensures consistent evaluation across different context management strategies while testing spatial reasoning and pathfinding capabilities.

**ALFWorld(ALF)**: ALFWorld (Shridhar et al., 2021) extends the TextWorld framework to household settings, requiring agents to navigate rooms and perform everyday activities involving commonsense reasoning. Tasks encompass diverse textual actions including object manipulation (picking up, placing items), furniture interaction, and environmental inspection. Each action undergoes validation against rule-based simulators providing textual feedback reflecting updated world states. Agent performance is evaluated using success rates, with episodes capped at 120 rounds and maximum full history length of 50. We employ the AgentGym library preserving the original system prompt to maintain consistency with established benchmarks while ensuring fair comparison across different context management approaches.

**TextCraft(TC)**: TextCraft is a text-only environment designed around Minecraft crafting mechanics, providing controlled settings for evaluating compositional reasoning and planning capabilities. The environment constructs crafting trees consisting of 544 nodes, each corresponding to craftable target items. For each task, environments specify target items alongside crafting com-

mand sequences derived from trees. Agents issue three action types: `craft <item> using <ingredients>`, `get <item>`, and `inventory`. Agents receive rewards of 1 only upon successfully crafting specified target items. Performance measurement uses success rates with maximum episode lengths capped at 80 steps. For full history experiments, maximum history windows are set to 60. We utilize the AgentGym library while adopting system prompts from (Prasad et al., 2023) with minor modifications to ensure compatibility.

**2048**: Game2048 appears in the TextArena suite (Guertler et al., 2025) as single-player logic puzzles adapted to text-based frameworks, originally inspired by sliding tile games. Tasks evaluate agents' abilities to reason over board states, tile merging mechanics, and long-term planning strategies. Environments simulate 4×4 grids where cells hold tile values as powers of two. Agents make moves in four directions causing tiles with identical values to merge into double-value tiles, thereby increasing game scores. We adapt evaluations for AgentGym framework compatibility while preserving TextArena's official scoring criteria with task sequences limited to 60 rounds and maximum history windows of 20 for full history strategies. Reward systems provide +1.0 for successfully reaching target tiles (default 2048). When games end without reaching targets, partial rewards are computed using weighted formulas: 50% based on score progress and 50

**FrozenLake(FL)**: FrozenLake adapts standard Frozen Lake grids from OpenAI Gym into TextArena's text-based framework. Agents navigate grids from start positions to goals ("G") while avoiding holes ("H") and traversing frozen tiles (empty spaces). Agent starting positions are marked as "P" and can begin from any corner. Actions correspond to four discrete directions with observations provided as visual text-based grid representations showing current board states with player positions. Unlike OpenAI Gym versions, our implementation excludes slippery surface mechanics—all movements are deterministic and execute exactly as commanded. Agents receive +1.0 rewards upon reaching goals. When falling into holes or exceeding step limits, partial rewards are calculated based on progress toward goals using BFS-computed shortest path distances. Episodes terminate upon reaching goals, falling into holes, or exceeding 40-round step limits. For full history experiments, maximum history windows are set to 20.

**Hangman(HM)**: Hangman implements classic word-guessing games within TextArena's text-based framework. Agents must deduce hidden English words by guessing individual letters or attempting complete words. Words are initially displayed as underscores ("_") with each underscore representing one letter. Game boards show column numbers (C00, C01, etc.) above letter positions for reference. Actions consist of single letter guesses formatted as "[L]" or complete word guesses as "[WORD]" (case-insensitive). When correct letters are guessed, all instances are revealed in respective positions. Agents maintain visible histories of previously guessed letters to avoid repetition. Observations include current board states showing revealed letters and underscores, remaining try counts, and already guessed letter sets. Reward systems provide +1.0 for successfully guessing complete words and partial rewards (0.0-1.0) calculated as ratios of correctly revealed letter positions to total word lengths when agents reach step limits. We adapt evaluations for AgentGym framework compatibility while preserving TextArena's official scoring criteria with task sequences limited to 40 rounds.

**RushHour(RH)**: Rush Hour is a classic sliding block puzzle game adapted as text-based environments for evaluating spatial reasoning and sequential planning abilities. Environments present 6×6 grid boards with vehicles of varying lengths (cars of length 2, trucks of length 3) positioned horizontally or vertically. Objectives involve maneuvering designated red cars (marked as 'X') to exits located at right board edges by sliding other vehicles out of paths. In each episode, agents interact with randomly generated puzzle configurations of varying difficulties (easy, medium, hard) determined by backward scrambling move numbers applied from solved states. Action spaces consist of vehicle movement commands formatted as [`vehicle_iddirection`], where `vehicle_id` represents capital letters (A-Z, with X representing red cars) and directions are '+' (forward/right for horizontal vehicles, down for vertical vehicles) or '-' (backward/left for horizontal vehicles, up for vertical vehicles). Vehicles move only along orientation axes and cannot overlap. Agents receive 1.0 rewards upon successfully guiding red cars to exit positions. For episodes reaching maximum step limits without solving puzzles, partial rewards are determined by red car proximity to exits. Performance is measured by success rates across puzzle difficulties with maximum episode lengths set to 50 steps and maximum history windows of 50 for full history strategies.

## O    ENVIRONMENT ANALYSIS AND METHOD GENERALIZABILITY

To validate our method's generalizability, we conduct experiments across diverse environment categories. Our evaluation encompasses embodied AI tasks (ALFWorld), web interaction (WebShop), reasoning-dependent strategic games (2048, Hangman, RushHour), navigation tasks (Maze, Frozen-Lake), and crafting games (TextCraft).

We categorize environments based on their dependency on multi-step decision-making: **Strong Multi-step Dependency** (Maze, WebShop) where environmental state transitions generate new observations requiring sequential reasoning; **Weak Multi-step Dependency** (TextCraft, ALFWorld) where some actions affect future states but with limited observation changes; and **Minimal Multi-step Dependency** (Hangman, 2048, RushHour, FrozenLake) where actions primarily depend on current state rather than extended interaction history.

Additionally, environments differ in failure modes: some allow quick failure recovery (most environments), while others like FrozenLake terminate immediately upon errors, limiting the impact of different context control methods since incorrect actions lead to rapid episode termination regardless of history management strategy.

Our experimental findings reveal that both training-free clip context and Context Preference Learning methods provide improvements across most environments. Notably, environments with strong multi-step dependencies (Maze, WebShop) show larger improvements, as these require models to maintain coherent understanding across multiple interaction rounds. In contrast, environments heavily dependent on reasoning capabilities show modest improvements, as our method addresses "how to understand multi-turn dialogue and maintain reasonable agentic exploration paradigms" rather than enhancing intrinsic reasoning abilities. Overall, our evaluation spans six distinct domains, three levels of sequential dependency, and multiple failure dynamics, offering broad coverage of agent interaction challenges and supporting the generalizability of our approach.

## P    PROMPT CONSTRUCTION

We structure multi-turn conversations using a sequential message format where each interaction cycle consists of system instructions, user goals, observations, and assistant actions. The conversation flow follows this pattern:

1. `system`: Environment-specific instructions and task descriptions

2. `user`: Initial goal or task specification

3. `user`: Current observation from environment

4. `assistant`: Generated action based on observation

5. `user`: Next observation after action execution

6. `assistant`: Subsequent action, and so on...

For observation formatting, we implement a two-part structure instead of direct text inclusion. Each observation is split into: (1) a structured header `user("Observation:\n")` followed by (2) the actual observation content `user(obs_content)`. This explicit formatting significantly improves Qwen3-8B's comprehension and task performance by providing clear semantic boundaries between different types of information.

# Q ATTENTION HEATMAP VISUALIZATIONS ACROSS ENVIRONMENTS

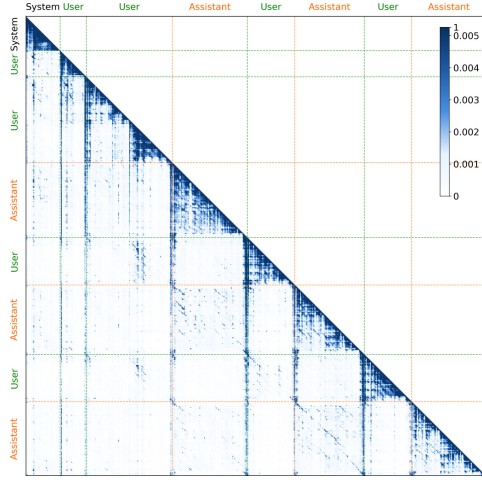

Figure 9: 2048 Environment Attention Patterns

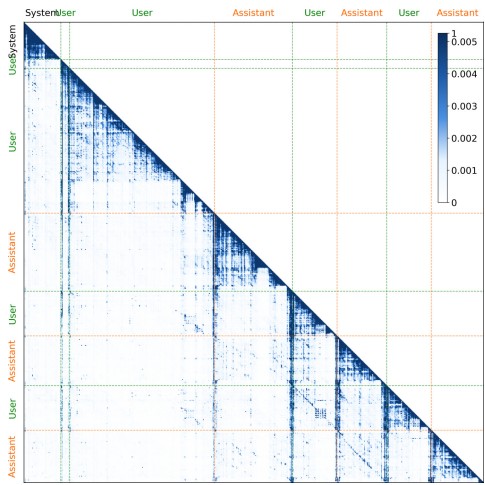

Figure 10: ALF Environment Attention Patterns

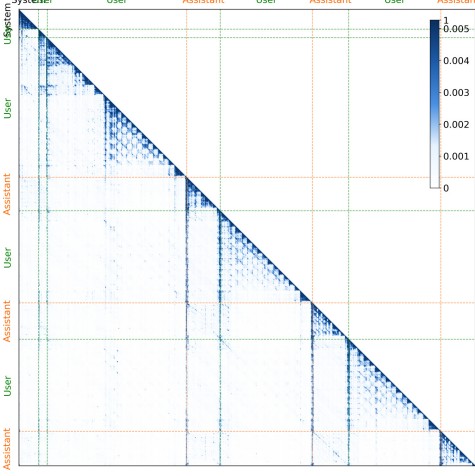
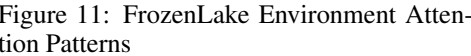

Figure 11: FrozenLake Environment Attention Patterns

Figure 12: Hangman Environment Attention Patterns

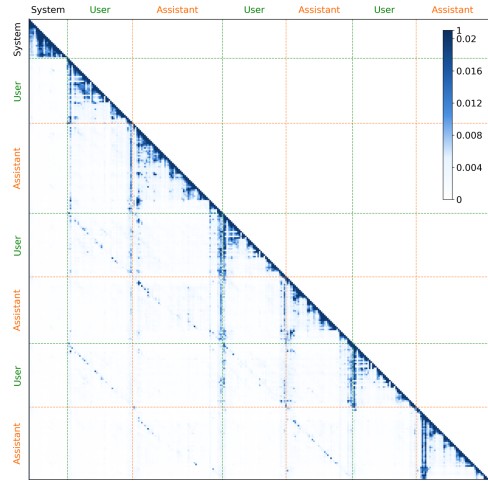

Figure 13: Maze Environment Attention Patterns

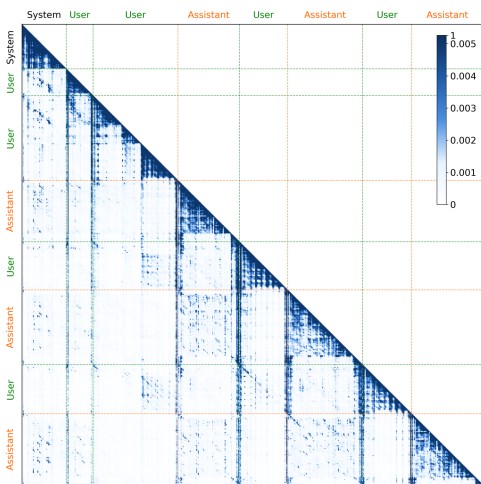

Figure 14: RushHour Environment Attention Patterns

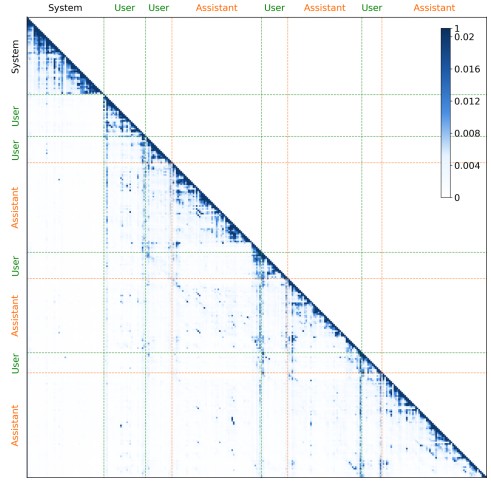

Figure 15: TextCraft Environment Attention Patterns

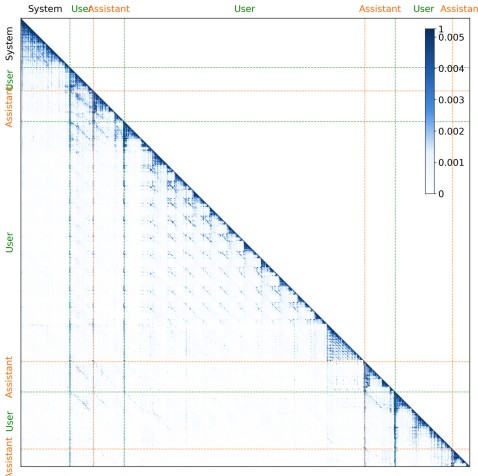

Figure 16: WebShop Environment Attention Patterns

# R   THE USE OF LARGE LANGUAGE MODELS

We use LLMs to search related works and identify relevant literature connections across the field of multi-turn dialogue agents and conversational inertia.

We use LLMs to aid and polish writing, including spell checking, finding appropriate vocabulary to express intended meanings, and generating initial drafts of paragraphs.

