# OpenReview forum: "Mitigating Conversational Inertia in Multi-Turn Agents through Context Bias Calibration"
_ICLR.cc/2026/Conference — Submitted to ICLR 2026_

### Official Review · Reviewer_1cYG · 2025-10-16

**Soundness:** 2
**Presentation:** 2
**Contribution:** 3
**Rating:** 4
**Confidence:** 4

**Summary:**

This paper addresses the performance degradation of multi-turn AI agents in long conversations , attributing the issue to a newly identified phenomenon called conversational inertia. This inertia stems from a strong diagonal attention pattern where the model excessively imitates its own previous responses, thereby suppressing exploration and accumulating errors. To solve this, the paper proposes a Context Bias Calibration framework featuring two complementary methods: a training-free clip context approach that periodically clears conversation history to break the inertial chain , and Context Preference Learning, which uses DPO to teach the model to prefer actions generated from shorter, less-inert contexts without needing environmental rewards. Experiments demonstrate that this framework effectively improves task success rates across eight diverse environments by fundamentally reducing the problematic diagonal attention and mitigating inertia.

**Strengths:**

1. This study investigates the performance degradation of multi-turn AI agents as conversations lengthen, attributing the root cause to a newly identified phenomenon called conversational inertia.

2. The training-free clip context method is simple, practical, and computationally efficient, notably preserving KV cache compatibility, which is a major advantage over sliding window approaches.

**Weaknesses:**

1. How can we be certain that reducing "diagonal attention" is the cause of the performance improvement, rather than merely a symptom of a better underlying strategy?

2. The paper treats "conversational inertia" as a negative bias. However, in certain tasks (e.g., following a multi-step, fixed procedure), imitating the format or behavioral pattern of the previous turn might be beneficial. Is there a risk that the proposed methods might incorrectly "calibrate away" useful behaviors in tasks where consistency is key?

3. The paper compares Clip Context with Full Context and Window Context. A stronger baseline might be context summarization techniques, where older conversation turns are compressed into a summary by an LLM.

**Questions:**

1. The paper defaults to H=12, L=1. How sensitive is the performance to these values? Is there a principled way to determine them, or is it purely empirical tuning?

2. Could there be scenarios where a shorter context lacks critical historical information, leading to a decision that appears exploratory but is actually naive? How does the method balance avoiding inertia against leveraging long-term history for well-informed decision-making?

3. Context Preference Learning fine-tunes the model to prefer "short-context thinking." Does this fine-tuning affect the model's capabilities on other, non-agentic tasks (e.g., general Q&A, text generation)?

---

> ### Author Response · Authors · 2025-11-20
> **Response to Reviewer 1cYG (Part 1/2)**
>
> Thank you very much for taking the time to review and for your support. We try our best to address your questions as follows.
>
> ## W1: Is reduced diagonal attention a cause or just a symptom?
>
> Our performance improvement is indeed achieved by reducing diagonal attention.
>
> **Reduced diagonal attention is the direct goal of DPO and the most significantly changed attention component after DPO.** The designed Context Preference Learning method favors strategies with low diagonal attention and high system attention (Fig. 2c) . The results naturally succeeded in reducing diagonal attention without rewards (Fig. 7), ultimately bringing performance improvements. Therefore, it can be considered a cause of poor performance, and improving diagonal attention can enhance performance. Notably, the goal is low diagonal attention and low assistant attention. After DPO training, assistant attention did not decrease as much. This further demonstrates that diagonal attention is an underlying variable that can be identified through this long-short comparison approach.
>
> ## W2: Does weak inertia degrade some conversation tasks?
>
> As shown in Figure 2, inertia can play either a negative or positive role in conversational tasks. In extremely restricted scenarios, we acknowledge that keeping moderate inertia is needed, but stronger inertia is not better because it will inevitably encounter the long context degradation problem.
>
> As the reviewer 3MMN notes: "you may find that ChatGPT gets stuck in some wrong directions ... you may start a new conversation and found that ChatGPT suddenly becomes smarter." In conversational agents, inertia is still something we don't want to be too heavy (Long context method). Only in extreme scenarios like fixed-format in-context learning for translation, keeping relatively more examples is helpful.
>
> What we claim is that directly applying LLM conversation patterns to multi-turn agent domains leads to performance degradation. Our method provides a simple, effective, and explainable approach for understanding what we meet when using conversational formats with LLMs.
>
> ## W3: Summarizer LLM as a baseline
>
> Summarization-based methods do not outperform Clip. We compared Clip with LLM-based summarization (ReSum) across both navigation environments (R-Table 1) and deep research scenarios (R-Table 2), finding that Clip achieves comparable or better performance while being simpler and more stable than summarization approaches that suffer from instability issues like context collapse and brevity bias. Please refer to General Response 1 and General Response 2 for detailed results.
>
>
> ## Q1: How sensitive is the performance to H and L values? Is there a principled way to determine them?
>
> **Performance sensitivity.** The Clip method shows lower sensitivity to hyperparameters compared to Window methods. As shown in General Response 3 (R-Table 5), when L=1, varying H from 2 to 12 shows consistent performance improvement (62.2→66.4→68.7→68.9), with the curve plateauing at H=12. This demonstrates that Clip is robust across a wide range of H values to outperform Window methods. R-Table 6 further reveals that Clip is robust across different L values. In contrast, Window methods show higher sensitivity: Window 1, 6, and 12 achieve scores of 62.2, 64.9, and 61.3 respectively, with performance degrading when the window is too large.
>
> **Principled selection guidelines.** We provide a principled approach based on task characteristics rather than pure empirical tuning:
>
> 1. **H parameter (upper limit):** Set H equal to or slightly higher than the typical window size W used in existing agent systems. H controls the maximum information available before clipping, so it should be large enough to accumulate sufficient decision-making context. Our experiments show H=12 works well across diverse environments.
>
> 2. **L parameter (lower limit):** The selection of L follows the principle of balancing exploration and exploitation based on task characteristics:
>    - For state-transition tasks (e.g., computer use), use low L (L=1 or L=3) to maintain strong exploration capability by reducing inertia.
>    - For reasoning-intensive tasks (e.g., deep research, multi-step problem solving), use moderate L (L=6) to retain sufficient reasoning history while still alleviating inertia.
>
> This principled approach makes Clip easy to apply: start with the window size already used in existing system as H, then adjust L based on whether the task primarily involves state transitions or reasoning.

---

> ### Author Response · Authors · 2025-11-20
> **Response to Reviewer 1cYG (Part 2/2)**
>
> ## Q2: Is the extra exploration from shorter context truly helpful, or just naive exploration from missing information?
>
> Strategies that "seem suboptimal in the current step" are essential for generalization in agentic tasks. The Clip method appropriately maintains such exploration diversity compared to Window methods. We explain this phenomenon from three perspectives: the role of controlled randomness, the inevitable limitations of long contexts, and the trade-off between information and exploration.
>
> **The role of controlled randomness in exploration.** The extra exploration is indeed helpful, though it operates through a mechanism that can be intuitively understood as analogous to (but fundamentally different from) temperature sampling. In agent tasks, including challenging scenarios like math reasoning, deterministic greedy decoding is rarely optimal. Instead, maintaining controlled randomness through temperature sampling enables exploration of alternative solution paths. Shorter contexts achieve a similar effect by reducing the model's over-reliance on historical patterns, thereby encouraging exploration of new strategies.
>
> **Longer context is not always better.** As shown in General Response 1, across all experiments, we consistently observe that performance under excessively long contexts is worse than under moderate-length contexts. From an empirical perspective, this demonstrates that the informational benefits of long contexts are outweighed by the disadvantages of reduced exploration diversity.
>
> **Missing information vs. reduced inertia.** While we acknowledge that shorter contexts do result in some information loss, our results show that the advantages of reduced inertia substantially outweigh this disadvantage. Current LLMs lack the ability to effectively utilize excessively long contexts without falling into repetitive patterns. In these cases, the inertia problem introduced by excessive historical information outweighs its informational benefits.
>
> ## Q3: Does Context Preference Learning affect the model's non-agent capabilities?
>
> Context Preference Learning preserves the model's original capabilities very well due to its reward-free design. We evaluated general capabilities and present them in R-Table 9:
>
> **R-Table 9: General capability evaluation**
>
> | Mode | Model | GPQA-Diamond (10 times) | MMLU-Redux |
> |---|---|---|---|
> | Non-thinking | Qwen3-8B | 48.83 | 79.13 |
> | Non-thinking | Qwen3-8B-DPO | 49.09 | 79.32 |
> | Thinking | Qwen3-8B | 58.03 | 83.04 |
> | Thinking | Qwen3-8B-DPO | 57.92 | 83.27 |
>
> This is because our Context Preference Learning method does not introduce bias as it is completely independent of rewards. We train on all trajectories, both correct and incorrect, without any LLM-informed verifier or human verifier. This makes this method nearly bias-free, which is rare among existing methods. Context Preference Learning does not damage general performance while effectively mitigating conversational inertia.

---

> > ### Comment · Reviewer_1cYG · 2025-11-23
> >
> > I appreciate the novel perspective on 'Conversational Inertia,' but I am concerned that the 'Clip Context' method is a brute-force heuristic that mitigates inertia by inducing amnesia without sufficient analysis of information loss. If the task is Partially Observable and a decision strictly relies on a clue observed $N$ turns ago (where $N > H$), this truncation strategy would theoretically guarantee failure. Could the authors discuss such failure cases where the loss of critical historical constraints outweighs the benefits of reduced inertia?

---

> ### Author Response · Authors · 2025-11-25
> **Response to Reviewer 1cYG**
>
> ## Discussion on information loss
>
> Thank you very much for your thoughtful and timely feedback. We provide detailed responses to your questions below.
>
> 1. **We acknowledge the information loss limitation and have provided further failure case analysis.** We follow your suggestion and have added a new section in Appendix L. We conducted controlled experiments on 50 randomly selected ALFWorld tasks and evaluated each task 16 independent times under Clip-6to1 configuration. We list failure cases where information loss outweighs inertia benefits below.
> 	- Sequential inspection tasks (Case 2435). These tasks require clear memory of previously checked items and current state to avoid redundant and repeated exploration. When the agent must inspect multiple locations sequentially, losing track of which locations have already been checked leads to repetitive actions and task failure.
> 	- Multi-step operations with strong dependencies (Case 2541). These tasks involve coherent multi-step operations where each action depends on outcomes from several turns prior. When context length falls below the dependency span, task success becomes nearly impossible. For example, if the agent must "pick up object A, go to location B, then use A on target C," losing the memory of having picked up A causes the agent to fail at the final step.
> 	- State-tracking tasks. These tasks require the agent to maintain awareness of environment state changes across multiple turns. Without sufficient history, the agent loses track of how its previous actions have modified the environment, leading to contradictory or redundant actions.
>
> Despite the above failure cases, we note that compared to Long context (which retains full history without any information loss), Clip-12to1 achieves a higher overall success rate on ALFWorld (Clip-12to1 achieves 67.7 while Long context achieves 61.0, as shown in Table 1). Additionally, we follow the same case study setting mentioned above and find that in tasks (e.g., Case 2531, 2425) with difficult non-intuitive operations, shorter context methods have less inertia and are less prone to being trapped in loops, thus having a higher probability of exploring correct directions. This demonstrates that while individual tasks may fail due to information loss, the overall performance benefits from reduced inertia.
>
> 2. **Information loss is an unsolved challenge across all context management approaches.**
> We emphasize that the information loss challenge is inherent to existing context management strategies, not unique to Clip.
> 	- The Window method drops conversation turns beyond the window boundary.
> 	- The summarization approach suffers from lossy and biased compression. As shown in Appendix K, summaries frequently exhibit over-claiming, premature conclusions, and missing critical information.
>
> Neither window methods nor summarization approaches adequately address the information loss problem. **Our Clip method does not aim to address the information loss challenge.** Instead, we identify and address a different problem (conversational inertia), which has been overlooked in prior work.
>
> 3. **Inertia reduction and information preservation can be addressed separately.**
> Here we discuss inertia reduction and information preservation in more detail. We agree that in some cases more information is valuable. However, our key insight is that inertia and information access are two different dimensions that can be addressed separately and combined effectively. R-Table 12 demonstrates this on BrowseComp.
>
> We can see that: (1) reducing inertia can outweigh information loss, as Clip-12to0 achieves better performance than Window-12 despite introducing some information loss by discarding history; (2) the information loss drawback can be recovered by combining Clip with summarization methods, where Sum-12to0 performs better than Clip-12to0; (3) alternatively, directly retaining some recent history as context also proves effective, as demonstrated by Clip-12to6 performing better than Clip-12to0.
>
> R-Table 12: Combining Clip (reduce inertia) with information-preserving methods in BrowseComp
>
> | Method | Score (+- SEM) |
> |---|---|
> | Window-12 | 24.2 +- 1.8 |
> | Clip-12to0 | 25.0 +- 1.8 |
> | Sum-12to0 | 27.7 +- 2.0 |
> | Clip-12to6 | 29.3 +- 1.5 |
>
> The above results imply that Clip's inertia reduction can be effectively combined with other information-preserving methods. Besides, the need to reduce inertia exists broadly in long-horizon multi-round agent scenarios: even a RAG-enhanced agent with task-relevant information retrieval still encounters conversational inertia problems when queries form multi-round interactions in its conversation context history. As a promising future direction, this inertia and information loss tradeoff could be mitigated by combining inertia-aware adaptive dropping with information-based retention, allowing critical earlier turns to be preserved while still reducing inertia.

---

### Official Review · Reviewer_3MMN · 2025-10-28

**Soundness:** 2
**Presentation:** 3
**Contribution:** 2
**Rating:** 2
**Confidence:** 4

**Summary:**

This paper studies a phenomenon in LLM called conversational inertia, in which the LLM will over-imitate the previous action and therefore limit its next-round exploration. By looking into the attention value, the author notices that this issue is caused by diagonal attention, that is, the LLM will pay more attention to the token that has the same relative position in the last conversation round. To tackle this issue, the author proposes context clipping and context preference learning to help the LLM focus more on the recent context. Their experiments covered a wide range of tasks, showing that context clipping and context preference tuning can solve conversational inertia effectively.

**Strengths:**

- The author provides a meaningful explanation for why the agent performs badly under a long context history, which is also a straightforward explanation for some of our daily tasks, e.g., you may find that ChatGPT gets stuck in some wrong directions in a multi-turn conversation and cannot always provide the expected answer to you, and you may start a new conversation and found that ChatGPT suddeny becomes smarter.
- The experiments show that the proposed method significantly outperforms the original long context LLMs.

**Weaknesses:**

- The proposed method is somehow naive and lacks novelty because people often deal with long context issues by summarizing the previous conversation rounds, but do not directly prune and discard them. But in this paper, the author does not involve such experiments. Also, the agent may be stuck in a loop of repeating early actions due to the lack of early-round memory. For some long-term tasks like deep research, such clipping will fatally affect the model's performance.
- On the other hand, similar designs of context management already exist in multi-agent systems, where they use an orchestrator to manage multiple agents, and each agent has their own context for subtask solving `[1, 2, 3]`. If the author thinks deeply about these multi-agent works, they may find that the context management in multi-agent works is quite similar to their proposed method (and even better):
    -  An orchestrator is designed to manage multiple subagents and solve a user task in multiple rounds
    - Each subagent has its own context.
    - Subagents will not keep memory from the previous term, which is identical to context clipping.
- The DPO loss design in this work has a very strong assumption, which is that the action generated from long-term history is worse than that generated from short-term history. I believe such a strong assumption only works on tasks that can be decomposed into multiple short-term subtasks, and the required steps for each subtask are less than or equal to the clipped threshold. I hope the author provides some ablation study in this part.

Refs:

[1] Chen, Weize, et al. "Agentverse: Facilitating multi-agent collaboration and exploring emergent behaviors." The Twelfth International Conference on Learning Representations. 2023.

[2] Song, Linxin, et al. "Adaptive in-conversation team building for language model agents." arXiv preprint arXiv:2405.19425 (2024).

[3] Zhuge, Mingchen, et al. "Language agents as optimizable graphs." arXiv preprint arXiv:2402.16823 (2024).

**Questions:**

- Somehow, I think the author has a very good discovery on why agents fail in multiple rounds, but their approach/solution becomes a regret of this paper. I would suggest the author focus more on how to solve the overwhelming instruction following issue based on what they've discovered (e.g., the diagonal attention) and try to keep a consistent agent memory when modifying the context.
- Table 1 does not look complete. Why are there no "Long" baselines and only "Window6" and "Clip12" for Llama-3-8B-instruct, gpt-4o-mini, and Qwen3-8B-RFT?

---

> ### Author Response · Authors · 2025-11-20
> **Response to Reviewer 3MMN (Part 1/2)**
>
> Thank you very much for taking the time to review and for your support. We try our best to address your questions as follows.
>
> ## W1: Can Clip be applied to deep search?
>
> Clip can be effectively applied to deep search scenarios by retaining moderate history.
>
> 1. Our method retains moderate historical information rather than clipping all the history. As shown in General Response 1, The purpose of clipping is not primarily to forget information, but to reduce inertia. In scenarios like deep research, which differ from the 8 simulated environments in the paper, retaining appropriate history after clipping works better.
>
> 2. As shown in General Response 2, model-generated summaries provide less accurate information than Clip method that directly uses recent history turns. The ReSum method (i.e., Sum 12to0) focuses only on summarization, but the results are not as good as directly retaining part of the trajectory, as shown in R-Table 2. The results of directly applying summaries from current single-agent base LLMs should be used with caution, as they have not demonstrated a consistently improving general approach.
>
> ## W2: The novelty of Clip Context - Simple, Effective, Generalizable
>
> We have cited these three papers in our revision and discussed them in the related work section. Compared to orchestrator-based approaches, Clip context's novelty lies in its simplicity, effectiveness, and broad generalizability across diverse environments without requiring environment-specific architecture.
>
> **Simple.** Clip context merely modifies one parameter of Window (General Response 3), enabling dropping multiple history turns at once without introducing any new models or additional prompts. Compared to orchestrator-based approaches, it is very easy to implement on existing agent frameworks. Window is currently the most widely used context management method.
>
> **Effective.** Clip context outperforms Window methods with performance improvements across multiple models, 8 simulated environments, and 1 deep research environment. Compared to summary methods, current models have not demonstrated sufficient summarization capability, and even the best summary methods show no improvement over Clip in these environments.
>
> **Generalizable:** Below we show three aspects demonstrating that Clip is more generalizable than orchestrator-based approaches.
>
> 1. **Easy to scale maximum steps.** Summary work suffers from summary accumulation explosion [3], while Clip behaves consistently at any step.
>
> 2. **Multi-environment scalability.** Current multi-agent frameworks often focus on only one type of scenario (e.g., Agentverse [6], AlphaEvolve [7], adaptive team-based systems [8]), meaning they have considerable environment-specific architecture and prompts. Real multi-turn scenarios include deep research, computer use, game playing, etc. These frameworks' effectiveness in broader environments has not been validated. Our method evaluates WebShop as computer use, RushHour as multi-step game playing, and BrowseComp as deep research, validating Clip's general effectiveness.
>
> 3. **Core difference: Prior knowledge.** Orchestrators appear to "work" in environments (e.g., MineCraft) because LLMs have prior knowledge of the task [9], enabling task decomposition so each subagent can perform "completable" and "predictable" subtasks. However, for tasks with unfamiliar operations (e.g., WebShop with unknown operations) or game playing (e.g., Maze), decomposition is impossible. Therefore, orchestrators cannot be applied to these unfamiliar scenarios. The Clip method, due to its simplicity, has no environmental priors and only improves single agents' exploration-exploitation balance.
>
>
> [3] Zhang, Q., Hu, C., Upasani, S., Ma, B., Hong, F., Kamanuru, V., Rainton, J., Wu, C., Ji, M., Li, H., Thakker, U., Zou, J., & Olukotun, K. (2025). Agentic Context Engineering: Evolving Contexts for Self-Improving Language Models. arXiv:2510.04618.
>
> [6] Chen, W., Su, Y., Zuo, J., Yang, C., Yuan, C., Qian, C., Chan, C.-M., Qin, Y., Lu, Y., Xie, R., et al. (2023). Agentverse: Facilitating multi-agent collaboration and exploring emergent behaviors in agents. arXiv:2308.10848.
>
> [7] Novikov, A., Vũ, N., Eisenberger, M., Dupont, E., Huang, P.-S., Wagner, A. Z., Shirobokov, S., Kozlovskii, B., Ruiz, F. J. R., Mehrabian, A., Kumar, M. P., See, A., Chaudhuri, S., Holland, G., Davies, A., Nowozin, S., Kohli, P., & Balog, M. (2025). AlphaEvolve: A coding agent for scientific and algorithmic discovery. arXiv:2506.13131.
>
> [8] Song, L., Liu, J., Zhang, J., Zhang, S., Luo, A., Wang, S., Wu, Q., & Wang, C. (2025). Adaptive In-conversation Team Building for Language Model Agents. arXiv:2405.19425.
>
> [9] Yu, S., & Lu, C. (2024). ADAM: An Embodied Causal Agent in Open-World Environments. arXiv:2410.22194.

---

> ### Author Response · Authors · 2025-11-20
> **Response to Reviewer 3MMN (Part 2/2)**
>
> ## W3: Wide applicability and effectiveness of the DPO assumption
>
> The DPO assumption is broadly applicable and empirically validated across diverse tasks and models.
>
> The assumption widely exists. Firstly, the assumption "Actions from longer contexts show stronger conversational inertia than those from shorter contexts" does not mean any longer context is worse than shorter contexts. Rather, it means excessively long contexts perform worse than moderate contexts. This assumption holds nearly universally across the evaluated 8 simulated environments, 1 deep research environment, and 3 types of models of various sizes and families, see Table 1 in revised paper. Related work [4] also found this assumption holds in indivisible tasks like Long-QA.
>
> If we don't follow this assumption, we will get lower performance, refer to R-Table 10. Using Qwen3-8B as base model, we evaluated Clip 12to1 with different preference pairs. We test two configurations: (1) preferring short over moderate contexts to naively add diversity, and (2) preferring moderate over excessively long contexts to alleviate inertia. Both improve performance, with the latter achieving the best results.
>
> **R-Table 10: Ablation on DPO preference pairs (Qwen3 base, Clip 12to1)**
>
> | Chosen | Reject | 8 env avg score |
> |---|---|---|
> | - | - | 68.9 (Base) |
> | 1 | 6 | 70.3 |
> | 6 | As long as possible | 72.5 |
>
> This method does not require tasks to be decomposable nor smaller than the H threshold. First, tasks like WebShop, 2048, and maze are completely non-decomposable and can only approach the answer through gradual state transitions. For environments without state transitions (deep research), we also show that excessively long contexts hurt final performance. Second, it does not depend on being smaller than the H threshold. In fact, Clip is better than Window holds across multiple mid-size settings as shown in R-Table 7. This has no relationship with the H threshold.
>
>
> ## Q1: Summarizer LLM as a baseline
>
> Summarization-based methods do not outperform Clip and show limited generalizability. Please see General Response 1 and General Response 2. In fact, we initially conducted summarizer-related experiments and found that current LLMs as summarizers show unsatisfactory performance and are not generalizable, which led us to deeply investigate the inertia problem brought by the simple clipping operation.
>
> ## Q2: Supplementing Long context as baseline
>
> We have supplemented experiments with additional models under Long context settings. Please see Table 1 in the revised paper for the complete comparison across different models. The Long context settings consistently perform worse than methods that keep moderate recent history, demonstrating that longer context is not always better.
>
> [4] Liu, N. F., Lin, K., Hewitt, J., Paranjape, A., Bevilacqua, M., Petroni, F., & Liang, P. (2023). Lost in the Middle: How Language Models Use Long Contexts. arXiv:2307.03172.

---

> ### Comment · Reviewer_3MMN · 2025-11-20
>
> Thanks for the response. I've check your rebuttal and the General Response 1, and here's my following reply to your response:
>
> > [In general response] Sum 12to1 68.8, Clip 12to1 68.9
>
> I think the gap between clip and sum is too small that I may consider such improvement comes from the error but not improvement. Could you also provide the breakdown of the scores (performance on each environment)? Also, could you try gpt-5-mini?
>
> > [In W2 rebuttal] Current multi-agent frameworks often focus on only one type of scenario (e.g., Agentverse [6], AlphaEvolve [7], adaptive team-based systems [8]), meaning they have considerable environment-specific architecture and prompts.
>
> > However, for tasks with unfamiliar operations (e.g., WebShop with unknown operations) or game playing (e.g., Maze), decomposition is impossible. ... The Clip method, due to its simplicity, has no environmental priors and only improves single agents' exploration-exploitation balance.
>
> These multi-agent systems do have some task specific prompts but your scenario have too. The main thing I want you to pay attention is their way to deal with long horizon tasks, which I think is similar to yours, i.e., clip the history for subtask agents, keep the history for the planner, but in your case, there is no planner. Also, could you clarify your second point? Subtask decomposition can also be treat as a back-and-forth exploration-exploitation approach.
>
> I would also increase my score as the author address part of my concerns.

---

> > ### Author Response · Authors · 2025-11-22
> > **Response to Reviewer 3MMN**
> >
> > Thank you very much for your thoughtful and timely feedback. In the following, we provide detailed responses to the key questions.
> >
> > ## Q3: Summarization performance breakdown and performance on GPT-5-mini
> >
> > Thank you for this suggestion. R-Table 11 shows the detailed performance breakdown across 8 environments for the qwen3-8B and GPT-5-mini models.
> >
> > **R-Table 11: Performance breakdown on qwen3-8B and GPT-5-mini**
> >
> > | Model | Context | MZ | ALF | WS | TC | FL | HM | 2048 | RH | Avg |
> > |-------|---------|----|----|----|----|----|----|------|----|----|
> > | Qwen3-8B | Clip-12 | 83.0 | 67.7 | 44.4 | 82.3 | 67.5 | 85.1 | 70.9 | 50.2 | 68.9 |
> > | Qwen3-8B | Sum-12 | 78.0 | 71.0 | 46.5 | 80.5 | 64.6 | 88.9 | 70.6 | 50.4 | 68.8 |
> > | GPT-5-mini | Clip-12 | 94.0 | 87.0 | 48.6 | 100 | 94.0 | 99.7 | 55.8 | 96.4 | 84.4 |
> > | GPT-5-mini | Sum-12 | 85.0 | 88.5 | 47.0 | 100 | 95.0 | 100 | 58.1 | 93.9 | 83.4 |
> >
> > The results show that summarization-based methods are comparable to Clip across diverse environments. Through our empirical case studies, we observe that one reason for the limited generalizability of summarization is its dependence on environment-specific prompts. In general settings, the model struggles to anticipate what information will be most useful for future steps, making it challenging to organize information optimally. Additional case studies are provided in Appendix K Example 4 to illustrate specific failure modes of summarization in general scenarios.
> >
> > ## Q4: The difference between orchestrator-based methods
> >
> > Thank you for this important question. We clarify the key differences between our approach and orchestrator-based methods below.
> >
> > **The fragility of decomposition, the importance of priors, and why orchestrator-based methods are scenario-limited:**
> > 1. Fragility of decomposition. Our point is more specific: task-specificity in existing multi-agent systems fundamentally stems from the decomposition schema itself—that is, planners and sub-agents are embedded within hand-crafted architectures that encode human priors about how tasks should be broken down. For example, AgentVerse[6] uses a Horizontal Structure specially designed for scenarios such as consulting or tool-use, particularly in its communication structures. This makes decomposition heavily environment-dependent and fragile.
> > 2. Importance of priors for decomposition. In a new environment—without pretrained model priors or human-crafted experience—systems tend to produce inefficient decompositions and plans. When the high-level decomposition is mis-specified, downstream agents will still faithfully execute a flawed plan. Empirically, this phenomenon has been repeatedly observed in multi-agent pipelines, which can underperform strong but simpler single-agent baselines due to coordination overhead and cascading errors [10, 11].
> > 3. **If proper decomposition timing/subgoals are missing, the orchestrator cannot leverage the exploration benefits that decomposition brings**. As discussed above, decomposition is crucial for the orchestrator, yet decomposition itself is scenario-fragile. Our method (planner-free) avoids this dependence. Specifically, our worker prompts directly adopt AgentGym's prompt format without injecting human-engineered task-specific decomposition schemas, making our approach more general.
> >
> > **We acknowledge that after decomposition, the early phase of each sub-agent resembles the effect of clipping history. The differences are:**
> > 1. The orchestrator still requires high-level exploration. As the central decision-maker, the orchestrator must also balance exploration and exploitation at the policy level. However, existing orchestrator methods mostly accumulate information and do not address this balance at the higher-level policy layer.
> > 2. Clip does not discard all history. By retaining only the most recent part of the context, Clip implicitly takes over the summarization role that the orchestrator would provide to each sub-agent—while being significantly simpler.
> > 3. Orthogonality. Clip provides a new mechanism that is orthogonal to and stackable with decomposition, and can be naturally applied to both sub-agents and the orchestrator. It specifically targets conversational inertia and imitation bias.
> >
> > **Clip is a strong baseline**. We are not claiming that Clip will always outperform summarization or the orchestrator in the future. Instead, we introduce Clip here as a strong and easy-to-implement baseline to highlight that:
> > 1. In general settings, Clip provides a more convenient and reliable starting point than orchestrator-based methods.
> > 2. LLM-based general summarizers even underperform Clip currently (General Response 3). We recommend that future work include Clip as a strong, easy-to-implement baseline to help disentangle whether improvements from summary or orchestrator methods come from better summarization quality or simply from the benefits Clip already provides.

---

> > > ### Author Response · Authors · 2025-11-22
> > > **Reference**
> > >
> > > [10] Barrak, A. (2025). Traceability and Accountability in Role-Specialized Multi-Agent LLM Pipelines. arXiv:2510.07614.
> > >
> > > [11] Cemri, M., Pan, M. Z., Yang, S., Agrawal, L. A., Chopra, B., Tiwari, R., Keutzer, K., Parameswaran, A., Klein, D., Ramchandran, K., Zaharia, M., Gonzalez, J. E., & Stoica, I. (2025). Why Do Multi-Agent LLM Systems Fail? arXiv:2503.13657.

---

> > > > ### Comment · Reviewer_3MMN · 2025-11-23
> > > >
> > > > I've read all the author's rebuttals. In general, I think the author has some interesting findings of why an LLM will fail to provide a correct solution, and proposes an easy and effective approach, context-clipping, to mitigate this issue. For an agentic task, as the environment status can play the memory's role in a multi-turn conversation, clipping context performs better than other methods like summarization and sliding window, but this method still has much room for improvement and still needs more analysis on other memory-sensitive tasks.
> > > >
> > > > Overall, I will raise my score to 6 for the interesting findings, and I hope the author can have more exploration on this topic.

---

> > > > > ### Author Response · Authors · 2025-11-24
> > > > >
> > > > > Thank you for your thoughtful follow-up and for raising your score. We appreciate your your insights on memory-sensitive tasks. We agree there is much room for further exploration, and we will continue to investigate broader settings in future work. Thank you again for your constructive feedback.

---

### Official Review · Reviewer_dcR6 · 2025-11-01

**Soundness:** 3
**Presentation:** 3
**Contribution:** 2
**Rating:** 4
**Confidence:** 3

**Summary:**

This paper identifies conversational inertia as a key failure mode in multi-turn LLM agents: with growing histories, models exhibit strong diagonal attention to prior assistant tokens, imitating their own earlier outputs and degrading performance even at moderate context lengths. The authors propose a Context Bias Calibration framework with two components. (1) Clip Context periodically clears interaction history, which both weakens the diagonal alignment and enables effective KV-cache reuse within a cycle. (2) Context Preference Learning performs DPO fine-tuning on long-short context action pairs, using the short context as input for both options to teach a preference for lower-inertia actions without environment rewards. Experimental results demonstrate that Context Bias Calibration framework reduces conversational inertia and achieves performance improvements.

**Strengths:**

1. The paper operationalizes “conversational inertia” via role-wise attention ratios and a diagonal-attention metric that reveal self-alignment to past assistant tokens, linking a visible attention pattern to a concrete failure mode and to targeted interventions.

2. The paper tests multiple base models across diverse environments, and pairs headline results with careful analyses, including role-wise/diagonal attention metrics, long-horizon scaling, “bad initialization” probes, and compute profiling.

3. Clip Context is training-free and Context Preference Learning is lightweight, so the method is easy to bolt onto existing agents while also delivering latency gains via better KV-cache reuse.

**Weaknesses:**

1. The proposed clipping scheme may underperform in environments that require uninterrupted short-term memory across turns. In the extreme, if the task is “sum of the last six observed numbers,” a Window-6 agent always retains the necessary evidence, whereas Clip-12 with L=1 periodically resets and cannot compute the target after resets because recent evidence is missing.

2. Based on the previous point, the paper needs to provide principled rules or diagnostics for selecting the clipping horizon based on task or environment. Otherwise, practitioners lack a reliable procedure to configure Clip for different environments.

3. Compared with the strong empirical and mechanistic support for Clip, the Context Preference Learning component offers thinner theoretical grounding and more limited ablations/comparisons (on only one model and one setting of context windows). The incremental gains are less thoroughly isolated from confounds, leaving its standalone contribution and generalization less clear.

**Questions:**

1. How should H and L be chosen? Are the strong results contingent on environments that are effectively restartable (i.e., the agent can clear all previous memory and start at any state)? I'm wondering what will happen if L increases.

2. Why is “same average input to agents” a fair basis of comparison? It does not align with compute complexity, context limits, or information volume, and inertia implies that “more rounds” can even be harmful. For example. could you justify this choice and additionally compare Clip-12 not only to Window-6 but also to bounds that bracket its effective context (e.g., Window-1 and Window-11)?

3. When is clipping appropriate across broader multi-turn tasks? Here are two examples. (a) For conversational agents with non-terminal feedback/rewards, periodic memory clearing may harm user experience (preference retention, discourse coherence). Is clipping still advisable? (b) In single-state tasks with ongoing feedback (e.g., coding/reasoning where the goal is fixed but observations change), does clipping effectively “reset and retry”? More generally, could you provide a taxonomy for when to use Clip vs Window vs full memory?

---

> ### Author Response · Authors · 2025-11-20
> **Response to Reviewer dcR6 (Part 1/2)**
>
> Thank you very much for taking the time to review and for your support. We try our best to address your questions as follows.
>
> ## W1: How does Clip context perform in environments that need uninterrupted memory access?
>
> Clip performs better than Window method in environments that need uninterrupted memory access by dynamically balancing exploration and exploitation through step-wise context variation.
>
>
> 1. **Clip achieves a balance between exploration and exploitation in long-turn scenarios.** We aim to solve a broad range of tasks across deep research, computer use, game playing, etc. To achieve this goal, the context management method should be robust. Though Window 6 can easily solve the mentioned task ("sum of the last six observed numbers"), it becomes problematic when the task is the sum of the last seven turns.  In contrast, the Clip method uses step-wise context length variation to solve this across multiple steps (see General Response 3). Since we cannot know in advance the maximum dependency limit, if we use Window 6 but the true requirement is the sum of the last seven turns, the problem can never be solved. The Clip method is more robust through its dynamic approach in **long-turn** evaluation.
>
> 2. **Why solving these tasks under short-horizon is not a good evaluation for agents.** We focus on scenarios with longer step limits in our paper. Because when the overall step limit is very small (e.g., 6), both Window 6 and Clip 12to1 degrade to full context, which is within the model's capacity. Besides, under small step limits, models must succeed on their first few attempts without sufficient exploration, and performance heavily depends on the model's prior familiarity with the environment (fitting degree) rather than examining the model's exploration and adaptation capabilities. Therefore, these short-turn evaluations may not comprehensively measure the model's true agentic capabilities.
>
> 3. **Clip context retains moderate historical information rather than clipping all of it.** As shown in General Response 2, in scenarios like deep research, "Clip 12to6" achieves a balance between retaining information and alleviating inertia. We did not search for this L=6 hyperparameter. A moderate parameter is sufficient and outperforms the Window method.
>
> ## W2 and Q1: How to select good parameters for Clip?
>
> Clip parameters can be easily selected based on task characteristics, with lower sensitivity compared to Window methods. As shown in General Response 3, Window context is a special case of Clip (Window W is completely equivalent to Clip L=W, H=W+1). When we decrease L while increasing H, the average input information remains constant, but the model's disruption of inertia becomes stronger when refreshing context.
>
> In practice, the H parameter of Clip can be kept at Window W or slightly higher. For the L parameter: in multi-turn scenarios where agent actions cause state transitions (e.g., navigation tasks), setting L to a lower value is sufficient (L=1, L=3). For tasks where each step involves equal reasoning (deep research or other multi-step reasoning), L needs to be set to a moderate value (L=6).
>
> Adding a new hyperparameter L does not increase the difficulty of practical application. Window context is widely used in agent systems, and existing approaches (e.g., UI-TARS[5]) already require hyperparameter tuning based on experience. We believe Clip's dual parameters do not impose greater tuning burden than Window methods. According to the experimental tables, a moderate Clip parameter is sufficient to outperform carefully selected Window parameters.
>
> [5] Qin, Y., Ye, Y., Fang, J., Wang, H., Liang, S., Tian, S., Zhang, J., Li, J., Li, Y., Huang, S., Zhong, W., Li, K., Yang, J., Miao, Y., Lin, W., Liu, L., Jiang, X., Ma, Q., Li, J., Xiao, X., Cai, K., Li, C., Zheng, Y., Jin, C., Li, C., Zhou, X., Wang, M., Chen, H., Li, Z., Yang, H., Liu, H., Lin, F., Peng, T., Liu, X., & Shi, G. (2025). UI-TARS: Pioneering Automated GUI Interaction with Native Agents. arXiv:2501.12326.

---

> ### Author Response · Authors · 2025-11-20
> **Response to Reviewer dcR6 (Part 2/2)**
>
> ## W3: Generalization and ablation analysis of Context Preference Learning
>
> Context Preference Learning is broadly applicable across different context management strategies and model families, with ablation studies confirming the importance of our core assumptions.
>
> We conducted additional experiments demonstrating that the DPO method works under multiple context management strategies (e.g., Clip, Window, and Long) and across two popular model families. We conducted additional Context Preference Learning experiments with Llama3.1-8B-Instruct using the same parameters as Qwen3-DPO in the paper (R-Table 8):
>
> **R-Table 8: Context Preference Learning results on Llama3.1**
>
> | Model | Context | 8 env avg score |
> |---|---|---|
> | Llama | Long | 47.1 |
> | Llama | Window | 55.3 |
> | Llama | Clip | 56.8 |
> | Llama DPO | Long | 48.2 |
> | Llama DPO | Window | 56.5 |
> | Llama DPO | Clip | 57.4 |
>
> We added ablation studies on long-short parameters, demonstrating the importance of our assumption that "excessively long context is worse than moderate context" to the results. We test two configurations: (1) preferring short over moderate contexts to naively add diversity, and (2) preferring moderate over excessively long contexts to alleviate inertia. Both improve performance, with the latter achieving the best results.
>
> **R-Table 10: Ablation on DPO preference pairs (Qwen3 base, Clip 12to1)**
>
> | Chosen | Reject | 8 env avg score |
> |---|---|---|
> | - | - | 68.9 (Base) |
> | 1 | 6 | 70.3 |
> | 6 | As long as possible | 72.5 |
>
>
>
> ## Q2: Why is Clip better than Window method?
>
> Clip outperforms Window because it dynamically balances the positive and negative aspects of context through periodic alternation between short and long contexts.
>
> As shown in General Response 3, we analyze the reasons for our parameter choices. As shown in General Response 1, the fundamental reason why Clip 12to1 outperforms Window 1 or Window 11 is that it balances both aspects of context effects.
>
> ## Q3: A taxonomy of context management methods
>
> Different task types require different context management strategies, with Clip being particularly effective for proposal generation and agentic tasks.
>
> We propose a taxonomy based on different task characteristics:
>
> **For proposal generation scenarios** (e.g., code agents like Claude Code), as turns increase, the diversity of proposed solutions decreases. To maintain solution diversity (otherwise stuck on one approach), it is necessary to alleviate the inertia problem through Clip.
>
> **For dialogue and information query scenarios**, the most effective approach is to write learned experience from history into the system prompt, converting it into independent queries rather than accumulating in conversation form. However, users are often unwilling or unable to comprehensively summarize their preferences and implicit requirements. Therefore, we discuss "how to manage context under this suboptimal usage":
>
> - For uncorrected forms by users, the model may mistakenly consider them as correct examples, so conversational inertia that Clip context aims to solve still exists in conversation scenarios, and Clip could remain superior to Window context.
> - Using Clip alone does lose some information (user habits), so using a model-based summarizer (e.g., OpenAI memory feature) as an auxiliary to Clip can be useful. The drawback is that summarizers need scenario-specific prompts and are quite sensitive and unstable.
>
> In practice, if there is a scenario-specific, strong summarizer (current LLM summarizers as single agents may not be good enough, see General Response 2), then we prefer summarizer-assisted Clip over Window and Full context. Otherwise, we prefer Clip over Window and Full context.

---

> > ### Author Response · Authors · 2025-11-26
> >
> > Dear Reviewer dcR6,
> >
> > We would appreciate it if you could let us know whether our clarifications address your concern. If there are any aspects that you feel would benefit from further explanation, we would be happy to elaborate.

---

### Official Review · Reviewer_Pp5L · 2025-11-02

**Soundness:** 2
**Presentation:** 3
**Contribution:** 2
**Rating:** 4
**Confidence:** 3

**Summary:**

This paper investigates why multi-turn dialogue agents degrade in performance at moderate context lengths (~ 4K tokens) despite modern LLMs' capacity for much longer sequences. The paper identifies "conversational inertia" where models exhibit strong diagonal attention patterns to previous assistant responses, essentially mimicking past actions rather than adapting to new environmental feedback. They proposed Context Bias Calibration framework addresses this through two mechanisms. 1st, training-free clip context method periodically truncates conversation history every H rounds to L recent rounds, breaking error propagation loops while enabling KV cache optimization for computational efficiency. 2nd, Context Preference Learning uses DPO to fine-tune few #params training models to favor actions generated from shorter contexts.

**Strengths:**

- Mechanistic evidence for conversation inertia is interesting and motivating where there is a clear degradation at moderate context lengths.

- Comprehensive experiments: Ablation experiments on the hyperparams used, different environments and a few models.

- DPO method doesnt require ground truth making the method more generalisable.

**Weaknesses:**

- While the conversation inertia is well motivated, I am struggling to see enough compelling evidence empirically. For example, the paper assumes diagonal attention causes poor performance, but one could observe similar pattern when there in an increase in the performance in the initial iteration.

- The solution probably works for short-term reasoning problems, where the agent "forgets" the history periodically. For longer tasks, the agent must actually "learn" from the experiences.

- I found some of the comparisions a bit unfair -Window-6 (always sees recent 6 rounds) vs Clip-12 (sees 1→12 rounds cycling).

- The paper claims that they introduced the concept of conversation inertia but I found many relevant papers that have not been cited:

[1] Hankache, Robert, et al. "Evaluating the Sensitivity of LLMs to Prior Context." arXiv preprint arXiv:2506.00069 (2025).

[2] Gupta, Akash, et al. "Llm task interference: An initial study on the impact of task-switch in conversational history." EMNLP (2024)

[3] Castillo-Bolado, David, et al. "Beyond Prompts: Dynamic Conversational Benchmarking of Large Language Models." Neurips D&B (2024).

**Questions:**

- A potential baseline: what if you ask LLM to summarise the past conversation?

- How would this method and hypothesis hold up in multi-step reasoning tasks?

- I can imagine these methods to have some variance, can you please report those.

- How does window-6 compare against clip-6?

---

> ### Author Response · Authors · 2025-11-20
> **Response to Reviewer Pp5L (Part 1/2)**
>
> Thank you very much for taking the time to review and for your support. We try our best to address your questions as follows.
>
> ## W1: Does conversational inertia appear in initial iteration?
>
> Conversational inertia does exist in initial iterations, but the positive aspects play the dominant role; only when rounds grow does it manifest as a major negative impact.
>
> To explain this, we first clarify the dual nature of context, with both positive and negative aspects. **Positive aspects:** Increased information enables more informed decision-making. **Negative aspects:** Introduces the inertia problem, which becomes increasingly severe as the number of turns grows.
>
> As the number of rounds increases, the intensity of conversational inertia gradually grows. While inertia exists even in initial iterations, the positive aspects of context dominate at that stage. Only after a sufficient number of turns does inertia emerge as the dominant drawback of context, manifesting as significant performance degradation. Therefore, both Window and Clip methods need to achieve a balance between these positive and negative aspects.
>
> ## W2: How does Clip Context work in long-term tasks?
>
> Clip context also works in long-term tasks by retaining moderate history rather than clipping all of it.
>
> As discussed in General Response 1, the purpose of the Clip method is not to forget, but to reduce inertia. Our method retains moderate historical information rather than clipping all of history. As demonstrated in General Response 2 on the long-term deep research task, retaining appropriate history after clipping (Clip 12to6) outperforms using only summary (Sum 12to0) or Window methods with strong inertia (Window 12).
>
> ## W3 and Q4: Fair comparison between Clip and Window methods
>
> Clip consistently outperforms Window methods in fair comparisons across various context lengths.
>
> Please see General Response 3. In R-Table 4, we observe the performance of Clip 7to1 is better than Window 6, and at the same time, Clip 13to1 is better than Window 12. This fair comparison shows Clip context is better than Window context, because of periodically reducing the conversational inertia when inertia becomes a strong negative factor.
>
> ## W4: Novelty of inertia
>
> We have cited these three papers in our revision and discussed them in the related work section. All these papers mainly focus on switching tasks in one conversation, which is turn-independent. In contrast, we mainly focus on indivisible turn-dependent tasks.
>
> Specifically, "Evaluating the Sensitivity of LLMs to Prior Context" found that relevant or irrelevant tasks can reduce performance on the current task. However, that task involved mid-conversation task switching where all prior information becomes noise for the LLM, which is not a proper use case but rather a special experimental scenario. "LLM task interference: An initial study on the impact of task-switch in conversational history" examined multiple task switches within a single conversation, where overall model performance degrades and fluctuates more significantly with increasing irrelevant context. "Beyond Prompts: Dynamic Conversational Benchmarking of Large Language Models" similarly studied multiple tasks in a single conversation, where LLMs easily confuse across tasks.
>
> Our Novelty on conversational inertia:
>
> 1. We focus on single coherent multi-step tasks and discover that inertia exists as an inherent flaw even when contexts are properly organized for LLMs, rather than focusing on task switching as a suboptimal usage pattern.
>
> 2. Previous work showed inertia only as confusion across multiple tasks. We discover that inertia reduces exploration ability even within the same task. Agent-related research (e.g., UI-TARS [5]) has not thoroughly investigated this continuous context performance degradation phenomenon and proposed a solution. Using Window Context, these works solve this problem mostly by hyperparameter search.
>
> 3. Beyond discovering the inertia phenomenon, we identify diagonal attention as a key cause of inertia, providing a deeper explanation. We validate through Context Preference Learning that reducing diagonal attention can mitigate inertia.
>
> 4. Prior work only showed that longer contexts lead to larger performance degradation. However, in agent environments, there is a trade-off between inertia and information, causing performance to first increase then decrease with context growth. These three works reveal interference phenomena but do not identify the opposing forces of beneficial information and detrimental inertia.
>
> [5] Qin, Y., Ye, Y., et al. (2025). UI-TARS: Pioneering Automated GUI Interaction with Native Agents. arXiv:2501.12326.

---

> ### Author Response · Authors · 2025-11-20
> **Response to Reviewer Pp5L (Part 2/2)**
>
> ## Q1: Summarizer LLM as a baseline
>
> Please refer to General Response 1 and General Response 2. These results show that Clip achieves comparable or better performance than summarization-based methods while being simpler and more stable.
>
> ## Q2: Applicability of the method and hypothesis to multi-step long-term reasoning tasks?
>
> Please refer to General Response 2. For multi-step long-term reasoning tasks that require multi-turn interaction with thinking before answering at each step, our findings and hypothesis fully hold.
>
> ## Q3: Variance on 8 environment results
>
> The average "within-case standard error of the mean (SEM)" across Qwen3-8B Window-6, Qwen3-8B Clip-12, and Qwen3-8B-DPO Clip-12 over 8 environments is about 0.5%, which is much smaller than the performance gap. The full "within-case SEM" calculation process and result are presented in Appendix H of the paper.

---

> > ### Author Response · Authors · 2025-11-26
> >
> > Dear Reviewer Pp5L,
> >
> > We would appreciate it if you could let us know whether our clarifications address your concern. If there are any aspects that you feel would benefit from further explanation, we would be happy to elaborate.

---

### Author Response · Authors · 2025-11-20
**General Response (Part 1/3)**

We sincerely thank all reviewers for their thorough reviews and constructive feedback. We are encouraged that the reviewers recognize our innovations in context management through the Context Bias Calibration Framework. We have carefully addressed all concerns raised and conducted additional experiments to strengthen our findings. In the following, we provide detailed responses to the key questions.

## General Response 1: Does clipping historical information lead to lower performance compared to baseline methods like summarization?

We appreciate this question about whether clipping historical context might degrade performance. To address this concern, we provide three key clarifications:

1. **Our method retains moderate historical information rather than clipping all of it.** The purpose of clipping is not primarily to forget information, but to reduce inertia. In our paper, we did not clip all history completely. Different environments may benefit from retaining different amounts of history. For the 8 evaluated task environments involving state transitions (navigation), we keep 1 context turn (Clip 12to1). For research scenarios discussed in General Response 2, we retain 6 context turns (Clip 12to6).  "Clip 12to1" means that we drop the 11 oldest history turns and keep 1 latest history turn when context rounds accumulate to 12. The corresponding hyperparameter is H=12, L=1.

2. **LLM-based summarization is comparable to our clipping method.** We conducted comprehensive comparisons with summarization methods, which are widely used in existing frameworks. We adopted the ReSum[1] prompt for summarization with a setting of Sum 12to1. "Sum 12to1" means based on Clip 12to1, when clipping, we add a summary of the current history. The summary buffer is updated when clipped again. We use Qwen3-8B as both actor and summarizer. More summary details can be found in Appendix I in the revision. Results averaged across 8 environments are shown in Rebuttal Table 1 (R-Table 1):

R-Table 1: Average performance across 8 environments

| Method | 8 env avg score |
|---|---|
| Long context | 54.5 |
| Window 6 | 64.9 |
| Sum 12to1 | 68.8 |
| Clip 12to1 | 68.9 |

The results show that Clip 12to1 achieves comparable performance to summarization while being simpler to implement. Clip does not need a summarization prompt, which can be sensitive to environments.
For comparisons between Clip and summarization in multi-step long-term reasoning environments, please see General Response 2.

3. **Clipping extends the principle that longer context is not always better.** While extending context can provide more information, it also introduces inertia that prevents agents from exploring new strategies. Our experiments (Table 1) demonstrate that Window significantly outperforms Long context by dropping history. Our Clip method drops larger chunks of history at once, which more effectively reduces this inertia compared to simple Window-based approaches, thus better leveraging the benefits of reducing inertia through history dropping.

[1] Wu, X., Li, K., Zhao, Y., Zhang, L., Ou, L., Yin, H., Zhang, Z., Yu, X., Zhang, D., Jiang, Y., Xie, P., Huang, F., Cheng, M., Wang, S., Cheng, H., & Zhou, J. (2025). ReSum: Unlocking Long-Horizon Search Intelligence via Context Summarization. arXiv:2509.13313.

---

> ### Author Response · Authors · 2025-11-20
> **Revised Paper**
>
> In the revised paper,
>
> 1. We updated Table 1 for the Long baseline, more Llama-3.1-8B-DPO model, and Qwen3-8B summary baseline
> 2. We added hyperparameter analysis and explanation of the Clip context method in Section 3.4
> 3. We added Section 3.6 to discuss the model's general capability after training
> 4. We added a comparison between the Clip context method and summary baseline in long-term reasoning tasks in Section 3.9
> 5. We added several related works discussions in Section 4
> 6. We added an explanation of how clipping oldest information can achieve greater performance in Section 5
> 7. We added an ablation study on Context Preference Learning parameters in Appendix C.2
> 8. We report the variance and SEM for these eight environments in Appendix H
> 9. We describe the summary implementation details in Appendix I
> 10. We describe the BrowseComp evaluation details in Appendix J
> 11. We added an empirical study of summary results in Appendix K

---

### Author Response · Authors · 2025-11-20
**General Response (Part 2/3)**

## General Response 2: Clip Context vs Summary Method on Multi-Step Long-Term Reasoning Scenarios (e.g., deep research)

We conducted experiments on multi-step reasoning scenarios to compare Clip and summary methods based on BrowseComp [2]. Proactive answer means the agent answers before the maximum step is reached, and forced answer means the agent answers only when the maximum step is reached. More evaluation details are provided in Appendix J of the paper. Results are shown in R-Table 2:

R-Table 2: Deep research scenario

| Method | Score (+- SEM) | Proactive answer rate | Proactive answer accuracy | Forced answer accuracy |
|---|---|---|---|---|
| Window 6 | 25.0 +- 1.7 | 27.3% | 55.7% | 13.4% |
| Window 9 | 23.4 +- 1.5 | 26.6% | 58.8% | 10.6% |
| Window 12 | 24.2 +- 1.8 | 25.4% | 58.5% | 12.5% |
| Clip 12to0 | 25.0 +- 1.8 | 29.3% | 56.0% | 12.2% |
| Sum 12to0 (ReSum) | 27.7 +- 2.0 | 63.7% | 37.4% | 10.6% |
| Clip 12to6 | 29.3 +- 1.5 | 30.9% | 61.0% | 15.1% |
| Sum 12to6 | 28.1 +- 1.6 | 30.8% | 63.3% | 13.2% |

These results further demonstrate the effectiveness of the Clip method:

- **Sum 12to0 (ReSum) performs worse than Clip 12to6:** Model-generated summaries provide less accurate information than directly using the 6 most recent history turns.
- **Window 6 performs worse than Clip 12to6:** Due to insufficient decision-making information, this degrades exploitation capability.
- **Window 12 performs worse than Clip 12to6:** Due to excessive inertia, this degrades exploration capability.
- **Window 9 performs worse than Clip 12to6:** Even though they have the same average input length, by perfectly balancing exploration and exploitation, Clip 12to6 achieves the best results.
- **Sum 12to6 is comparable to Clip 12to6:** When historical examples are already available, adding an extra summarizer does not bring improvements.

Existing summarization approaches show unstable performance, having Context Collapse and Brevity Bias problems [3]. Through our empirical study, we also found issues with summary-based methods such as over-confidence (e.g., model-generated summaries may lead the actor model to have higher proactive answer rates but lower answer accuracy). More detailed empirical analysis can be found in Appendix K in the revision. Therefore, our method does not adopt summarization. In contrast, our Clip method is simple as an applicable approach, and we recommend it as a strong baseline.


[2] BrowseComp: A Multi-Step Web Browsing Benchmark. OpenAI. https://openai.com/index/browsecomp/

[3] Zhang, Q., Hu, C., Upasani, S., Ma, B., Hong, F., Kamanuru, V., Rainton, J., Wu, C., Ji, M., Li, H., Thakker, U., Zou, J., & Olukotun, K. (2025). Agentic Context Engineering: Evolving Contexts for Self-Improving Language Models. arXiv:2510.04618.

---

### Author Response · Authors · 2025-11-20
**General Response (Part 3/3)**

## General Response 3: A Fair Comparison Between Clip and Window

We note that Window is a special case of Clip (e.g., Window 6 equals to Clip 7to6). To provide a comprehensive comparison, we conducted extensive ablation studies on Clip parameters (R-Table 4).

R-Table 4: Clip parameter ablation

|  | H=2 | H=3 | H=6 | H=7 | H=12 | H=13 |
|---|---|---|---|---|---|---|
| L=1 | 62.2 (W=1) | 66.4 | 68.7 | - | 68.9 | - |
| L=6 | - | - | - | 64.9 (W=6) | 62.1 | - |
| L=12 | - | - | - | - | - | 61.3 (W=12) |

Below we analyze Clip parameters and compare them with Window from three perspectives:

1. **Effect of H parameter.** The upper limit H provides more contextual information for decision-making, making the agent more informed when proactively answering. R-Table 5 shows the effect when L=1:

R-Table 5: Effect of H parameter (L=1)

| L=1, H=? | L=1, H=2 | L=1, H=3 | L=1, H=6 | L=1, H=12 |
|---|---|---|---|---|
| 8 env avg Score | 62.2 | 66.4 | 68.7 | 68.9 |

2. **Effect of L parameter.** The lower limit L controls the ability to break inertia: lower L values strengthen the ability to propose new methods and paths, making it more suitable for exploratory tasks; moderate L values retain more information, making it more suitable for summarization and search tasks. R-Table 6 shows the effect when we mainly adjust L:

R-Table 6: Effect of L parameter

| hyperparameter | L=1, H=12 | L=3, H=12 | L=6, H=12 | L=11, H=12 |
|---|---|---|---|---|
| 8 env avg Score | 68.9 | 67.1 | 62.1 | 60.7 |


3. **Clip vs Window.** When Clip and Window method have the same average input to model, Clip (L=1) outperforms Window at various context lengths (R-Table 7):

R-Table 7: Clip vs Window at different context lengths

| Hyper-parameter| Clip (L=1,H=?) | Window (W=?) |
|---|---|---|
| L+H=7,W=3 | 68.7 | 64.0 |
| L+H=11,W=5 | 68.8 | 64.2 |
| L+H=13,W=6 | 68.9 | 64.9 |
| L+H=15,W=7 | 68.9 | 63.8 |

From the above experiments, we can analyze the advantages of Clip over Window:

1. **Superior performance compared to Window context.** We fairly compared Clip with Window from multiple perspectives: R-Table 7 shows that with optimal parameters and various context lengths, Clip consistently outperforms Window methods. R-Table 5 demonstrates that with the same H (memory limit), Clip progressively outperforms Window methods as inertia is reduced. Our Clip method periodically alternates between short contexts (for exploration by reducing inertia) and long contexts (for exploitation by utilizing interaction history), achieving a dynamic balance across conversation turns. We adopted the approach of balancing average input tokens ($L=1, H=2*W$) as a comparison described in the paper.

2. **Lower hyperparameter sensitivity.** Referring to R-Table 5 and 7, compared to Window, Clip is less sensitive to hyperparameters and does not introduce difficult parameter tuning. This is also because the Clip method achieves dynamic balance across conversation turns. This design makes Clip adaptable to scenarios with different requirements.

---

### Meta-Review · Area_Chair_1Xdr · 2026-01-01

**Summary:**

The review process primarily focused on concerns regarding the lack of comparison against standard summarization baselines, the potential for critical information loss in long-horizon reasoning tasks due to the proposed context clipping mechanism, and the robustness of the hyperparameter selection.

**Reviewer Concerns:**

The rebuttal effectively addressed the concerns regarding summarization baselines and information loss through new comparative experiments (including BrowseComp) and ablation studies, effectively resolving the major technical reservations regarding the method's robustness and comparative performance.

**Reviewer Scores:**

Do not anticipate score raise; the unaddressed flaws in the evaluation protocol substantially undermine the paper's claims of improvement over existing methods.

---

### Decision · Program_Chairs · 2026-01-26

Reject